# Dual-task costs of listening while driving in older and younger adults

Katherine Bak[1,2], Kristen Arnold[1], Lena Darakjian[1], M. Kathleen Pichora-Fuller[2], Frank A. Russo[1,3], Jennifer L. Campos[1,2]*

1 KITE - Toronto Rehabilitation Institute, University Health Network, Toronto, Ontario, Canada,
2 Department of Psychology, University of Toronto, Toronto, Ontario, Canada,
3 Department of Psychology, Toronto Metropolitan University, Toronto, Ontario, Canada

* Jennifer.Campos@uhn.ca

## Abstract

Age-related changes in hearing and cognition result in increased listening difficulties, which could affect the ability to perform common, complex, multitasking behaviours, such as listening while driving. However, there are very few realistic and controlled studies examining how the competing attentional demands of listening while driving affect performance and how performance may differ between younger and older adults. The primary objective of this study was to examine dual-task costs of listening while driving in older ($N = 24$, $M_{age} = 68.29$ years, 15 female) and younger ($N = 24$, $M_{age} = 26.42$ years, 12 female) licensed drivers with normal hearing, vision, and cognition. Participants completed a driving task in a high-fidelity driving simulator during simpler (Rural driving) and more complex (City driving) driving conditions. They also completed the Connected Speech Test (CST) at +4 signal-to-noise ratio (SNR; easier listening) and at 0 SNR (harder listening) conditions. Finally, they completed both tasks simultaneously to examine the dual-task costs of listening while driving. Results demonstrated that only older adults showed significantly poorer listening accuracy on the CST in the dual-task compared to the single-task condition, particularly during the more difficult driving conditions (City driving). Both older and younger adults showed poorer driving performance (greater variability in lane position) in the dual-task compared to the single-task condition, which was most pronounced under the most challenging conditions (City driving, 0 dB SNR listening). However, the magnitude of these dual-task costs during the most challenging conditions was greater in older adults than younger adults. Findings may inform mitigation strategies to reduce the effects of difficult listening conditions on driving performance by optimizing vehicle acoustics or by minimizing auditory distractions, particularly during more challenging driving conditions.

**Data availability statement:** Data cannot be shared publicly because of strict data sharing policies at the University Health Network (UHN). Data access may be granted from the UHN Ethics Committee (contact via reb@ uhnresearch.ca) for researchers who meet the criteria for access to confidential data.

**Funding:** This research was funded by a Natural Sciences and Engineering Research Council of Canada (NSERC) Discovery Grant (RGPIN-2021-03453) awarded to JLC and an NSERC Canada Postgraduate Scholarship- Doctoral (568833 - 2022) to KB. JLC also holds a Canada Research Chair II in Multisensory Integration and Aging (950-232488) from the Government of Canada. https://www.nserc-crsng.gc.ca/index_eng.asp https://www.chairs-chaires.gc.ca/home-accueil-eng.aspxSponsors/funders played no role.

**Competing interests:** The authors have declared that no competing interests exist.

## Introduction

Listening during complex, multisensory, multitasking conditions can be difficult for older adults due to age-related changes in auditory and cognitive abilities. These changes can include: elevated auditory pure-tone thresholds and reduced auditory processing speed and accuracy, working memory, inhibitory control, divided attention capacity, and task switching/planning [1–7]. These age-related changes may lead to increased listening difficulties, especially in noisy conditions, requiring more cognitive resources to listen, thereby reducing the spare cognitive capacity available to support other simultaneously performed tasks [6].

One common everyday behaviour during which individuals are required to listen while simultaneously performing a complex task, is driving. Despite it seeming to be automatic, driving is a highly complex task, requiring multisensory processing, divided attention, and motor control. Drivers often listen to the radio, audiobooks, converse with a passenger, or on their cell phone while driving. The strategic allocation of perceptual, cognitive, and motor resources among the various processes involved in listening and driving tasks is critical, particularly when limited resources and/or non-optimal allocation of resources may lead to performance decrements and compromise safety.

Allocation of cognitive resources can be examined using dual-task paradigms. In the context of studies on listening effort, dual-task paradigms often introduce a speech recognition task (e.g., word recognition in noise), with a competing secondary task, such as a visual detection, visual attention, or memory task [8–10]. Single-task performance is subtracted from dual-task performance to measure dual-task costs. If processing resources are spent on listening, spare cognitive capacity may be reduced, thereby leading to poorer performance on one or both tasks, depending on task prioritization. If the resources required to succeed on both tasks do not exceed available cognitive capacity, dual-task performance may closely approximate single-task performance (i.e., a difference of zero between single- and dual-task performance). Most studies have found that older adults experience greater listening effort during speech-in-noise tasks compared to younger adults, evidenced by poorer secondary task performance during dual- compared to single-task conditions [8–10]. Studies of listening effort often test participants in controlled, isolated environments (e.g., sound booths), using artificial secondary tasks and stimuli, thereby lacking ecological validity [11,12]. Far less is understood about how the allocation of processing resources toward listening influences performance during realistic, everyday multitasking activities that involve listening.

The complex task of driving reflects real-world acoustical challenges (e.g., background noises, reverberation), other challenging environmental conditions (e.g., poor weather) and perceptual challenges to other sensory modalities (e.g., reduced visual inputs at night), and requires coordinating multiple sensory, cognitive, and motor tasks simultaneously. Therefore, a "listening while driving" dual-task paradigm can provide insights into cognitive resource allocation and task prioritization in older and younger adults during more ecological valid situations. Other research in the context of gait-related mobility has shown that when coordinating the dual-task requirements

of listening while walking/balancing, older adults often prioritize safety (e.g., walking without falling) over other simultaneously performed perceptual or cognitive tasks such as listening [13–15]. Driving is another important example of a safety-critical, mobility-related task that depends on strategic resource allocation, which may change with older age.

Broader research investigating the effects of distracted driving or dual-task driving has used relatively simple driving scenarios. For example, previous driving simulation and closed-road circuit studies often include basic driving tasks (e.g., drive straight, brake when lead car brakes) and simple visual environments (e.g., grasslands, rural road with few buildings), which may place less demand on cognitive, perceptual, and motor resources compared to more complex driving conditions. Consequently, these types of scenarios likely underestimate the allocation of perceptual, cognitive, and motor resources during real-world driving challenges.

Most previous studies that implemented an auditory task within a dual-task driving paradigm were primarily interested in the effects of distraction and/or cognitive demand broadly (e.g., talking on a cell phone while driving), rather than the effects of auditory processing and listening per se. Auditory distraction tasks have included, for example, repeating auditorily-presented digits, performing math operations, completing auditory n-back tasks, or Go-No-Go tasks, and listening to and remembering traffic news segments [16–19]. Overall, such studies have generally reported no age-related differences in dual-task costs of performing an auditory task while driving on driving performance (but see [20]). These null effects of age may be partially attributable to the driving tasks not being sufficiently challenging, and/or the auditory tasks not strategically targeting domains of auditory functioning that are known specifically to decline with older age (e.g., auditory processing speed and accuracy, working memory, inhibitory control). Usually, auditory task characteristics have not been strategically manipulated to allow for the examination of how age-related differences in auditory processing may influence driving (e.g., manipulations to signal-to-noise ratios, availability of contextual support, linguistic complexity). Speech-in-noise tasks may be more appropriate for targeting auditory processing and may be more sensitive to known age-related changes in sensory and cognitive abilities than the other types of basic auditory tasks typically used in past dual-task driving studies. Previous literature has shown associations between hearing and cognition in healthy older adults [21–23]. However, to our knowledge few experimental studies have examined how hearing measures (i.e., pure-tone threshold average [PTA], words-in-noise test scores) and cognitive measures (e.g., working memory, executive control) are independently associated with performance on more complex listening-while-driving dual-tasks.

Although little is known about how auditory processing demands affect listening while driving in older adults with normal hearing, there is some evidence from epidemiological and self-report studies on the relationship between hearing loss and real-world driving outcomes showing mixed results [24–28]. With respect to lab-based research, to our knowledge, at least two studies have examined the effects of auditory task performance on driving in older adults with age-related hearing loss [29,30]. Results from these studies demonstrated dual-task costs to driving performance under certain driving conditions in older adults with moderate to severe hearing loss compared to older adults with mild to no hearing loss [29] and dual-task costs to both listening and driving performance in older adults with hearing loss under aided and unaided conditions [30]. While there is some evidence that clinically significant hearing impairments may negatively affect listening and driving performance, it is unclear if similar effects are observed for older adults with common supra-threshold declines in hearing; in other words, when speech is audible but difficult to understand, particularly in noise, due to age-related changes to cognitive and auditory processing [31].

## Current study

The primary objective of the current study was to examine dual-task costs of listening while driving in younger and older adults with clinically normal hearing using a high-fidelity driving simulator. Further, we also considered the effects of higher vs. lower listening and driving demands on dual-task costs to listening and driving performance. For the listening task, we used the Connected Speech Test [32], a complex, realistic, and standardized conversational-style speech-in-noise task. Two different signal-to-noise ratios were used to manipulate listening difficulty and examine the effects of higher vs. lower

auditory processing demands on driving performance. For the driving task, both simpler driving scenario elements (e.g., rural roads) and more complex scenario elements (e.g., city roads) were used to examine the effects of driving difficulty on task performance and dual-task costs. It was hypothesized that older adults would show greater dual-task costs to listening and driving performance compared to younger adults, particularly in the more challenging listening and driving conditions. It was also hypothesized that when both listening and driving difficulty were high, participants would be less able to coordinate both tasks, resulting in greater relative costs to listening compared to driving, particularly for older adults. A secondary goal of the current study was to explore associations among listening and driving performance and measures of hearing and cognition.

## Methods

### Participants

Twenty-eight younger adults and 33 older adults participated in the study. Four younger adults and 9 older adults withdrew due to motion sickness. For participants who completed the entire study, ratings of motion sickness were very low (see details below). Thus, the final dataset included 24 younger adults ($M_{age}$ = 26.42 years, Range 20–36; 12 Female, 12 Male) and 24 older adults ($M_{age}$ = 68.29 years, Range 61–80; 15 Female, 9 Male). All participants provided written informed consent, spoke fluent English, learned English by the age of 5, reported English as their primary language of choice (with the exception of one younger adult and one older adult participant), had a valid driver's license with at least two years of driving experience, reported no history of any major health conditions, did not wear hearing aids, and had normal hearing, vision, and cognition (see below for details on specific eligibility criteria). While nearly all participants completed every baseline assessment described below, some participants did not complete certain tests (unrelated to eligibility). See S1 Table for total *N*'s for each test and group. Participants were compensated $20 per hour. The study was approved by the University Health Network (REB 19–5957) Research Ethics Board. Data collection started July 7, 2022, and ended February 5, 2024.

### Baseline assessments

**Hearing.** A screening hearing test was administered in-person using Shoebox™ (Version 5.6.7); participants' pure-tone audiometric air-conduction thresholds (dB HL) were measured in each ear across five frequencies of 500 Hz, 1000 Hz, 2000 Hz, 3000 Hz, and 4000 Hz to determine eligibility (Fig 1). All participants had clinically normal hearing, defined as an average of 25 dB HL or less in the better ear across the five frequencies tested [criteria adapted from 33,34], apart from one younger adult whose audiometric data did not save due to technical issues but this participant self-reported normal hearing. All participants also had no significant interaural asymmetries (defined as >15-dB interaural difference in pure-tone threshold average (PTA), [35]), apart from two older adult participants who had 17 dB HL and 19 dB HL interaural difference in PTA. Additional measures of hearing were collected in person to characterize behavioural and self-reported hearing abilities, respectively the Canadian Digit Triplet Test (CDTT) [36] is a speech-in-noise test that measures speech reception thresholds in noise and the Hearing Handicap Inventory – Short (HHIE-S) questionnaire [37]. These additional measures were not used to determine eligibility but rather to examine exploratory associations with primary outcome measures. The two participants who had interaural differences between PTA scores >15 dB HL performed similarly on these additional auditory measures and on the main experimental task, compared to participants with <15 dB interaural difference in PTA scores. Therefore, these two participants with greater interaural differences did not likely skew the data and we decided to include them in the analyses to increase statistical power.

**Vision.** The Early Treatment Diabetic Retinopathy Study (ETDRS) [38] eye chart was used to characterize far visual acuity in the left and right eyes separately. All participant's visual acuity in each eye was below the clinically normal cut-off of ≤ 0.3 logMar units, except for one older adult participant who had mild vision loss of ≥ 0.48 logMar units in both eyes [39]. Participants wore corrective lenses during the acuity test if they also wore them during the experiment.

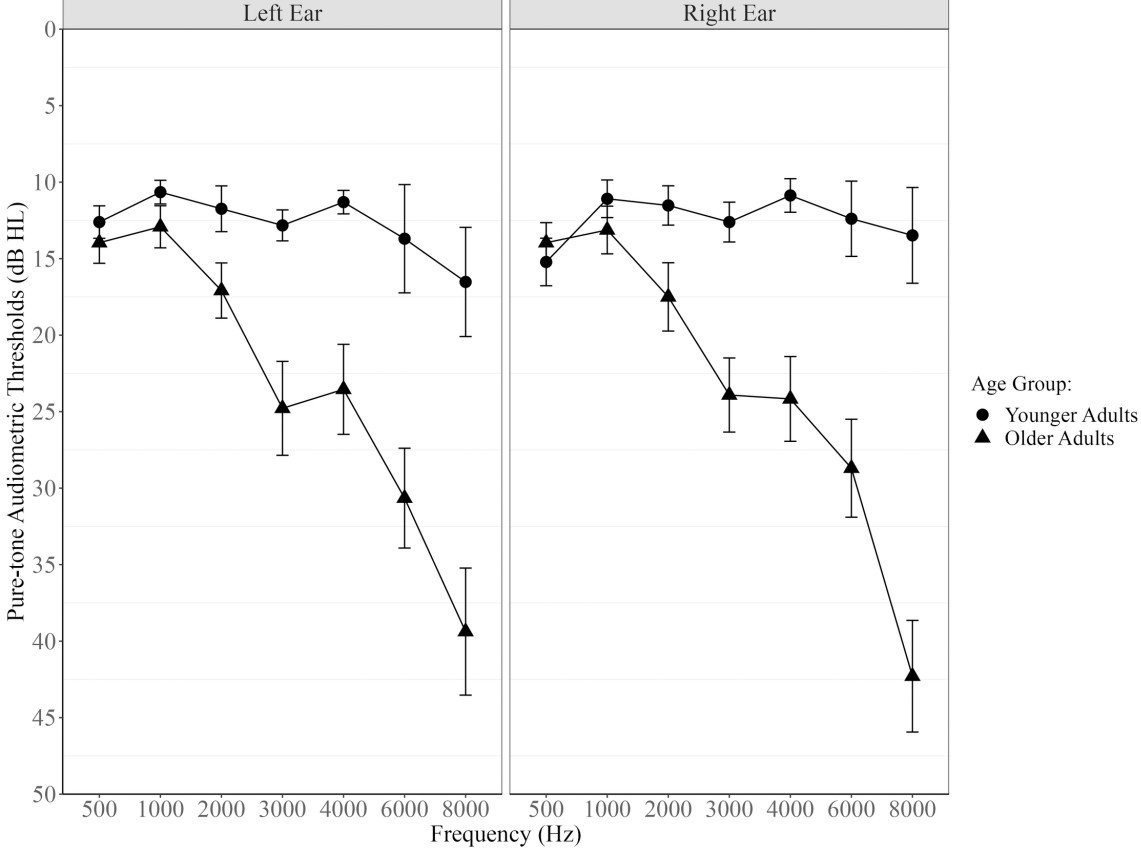

**Fig 1. Hearing thresholds at each frequency (500 Hz – 8000 Hz) and ear for both younger and older adult groups.** The circle and triangle symbols represent the means. Error bars represent ±1 SE.

**Cognition.** The Montreal Cognitive Assessment (MoCA) was administered to all older adult participants to screen for mild cognitive impairment (MCI) [40]. The initial recommendation of 26/30 as the cut-off score for MCI [40] may lead to a high false positive rate for MCI than a cut-off score of 23/30 [41]. All participants scored at least 23/30, except one participant who scored 21/30. This participant's performance on other cognitive measures (see below) was poorer than Canadian normed scores for their age; however, their data were retained because their main experimental data were not outliers. Other baseline measures of cognition included The Trail Making Tests A and B to measure planning, processing speed, and task-switching [42], the Stroop test to measure inhibitory control [43], and the Digit Span Forward, Backward, and Sequencing test to measure working memory capacity [44]; these cognitive measures were not used to determine eligibility but only to examine exploratory associations with primary outcome measures.

The Useful Field of View (UFOV) test measured participants' ability to attend to objects in their central visual field while also attending to objects in parts of their periphery using three subtests of, 1) Processing Speed of the object in the central visual field only, 2) Divided Attention between objects in the central visual field and the periphery, and 3) Selective Attention for objects in the central visual field and the periphery while ignoring irrelevant visual distractors [45]. The UFOV was not used to determine participant eligibility but rather to examine exploratory associations with primary outcome measures. See Table 1 for comparisons between groups for each baseline measure.

## Self-reported demographic questionnaires

Participants completed an in-house Health History Questionnaire (HHQ) which asked various health-related questions and a Driving Habits Questionnaire (DHQ) (adapted from [46]) which asked various driving-related questions. The HHQ and DHQ were not used to determine eligibility to participate in the study but rather to better characterize participants.

## Self-reported motion sickness

Participants completed a Motion Sickness Susceptibility Questionnaire (MSSQ-Short, [47]) to better characterize participants and be aware of those at an increased risk of developing simulator sickness; the measure was not used to determine eligibility for the study. Participants were also asked to rate their feelings of motion sickness using the Fast Motion Sickness (FMS) scale ranging from 0 (no motion sickness) to 20 (severe motion sickness to the brink of vomiting) throughout the study [48]. More details regarding the FMS are described below under the Procedure section.

**Table 1. Participant group demographics and baseline assessment outcome measures.**

| | Older Adults | | Younger Adults | | |
|---|---|---|---|---|---|
| | *M* | *SD* | *M* | *SD* | *t*(df) |
| **Demographics** | | | | | |
| Age (years) | 68.29 | 4.76 | 26.42 | 3.40 | 35.08(46)*** |
| Education (years) | 16.46 | 2.36 | 17.54 | 2.43 | -1.57(46) |
| Driving Experience (years) | 49.05 | 7.58 | 9.88 | 8.96 | 15.94(44)*** |
| Self-Reported Driving Rating[1] | 4.23 | 0.75 | 4.17 | 0.70 | 0.28(44) |
| **Hearing** | | | | | |
| PTA better ear (dB HL) | 15.84 | 5.38 | 11.13 | 3.91 | 3.42(45)** |
| CDTT SRT (dB SNR) | -11.13 | 1.06 | -11.93 | 0.71 | 3.07(46)** |
| **Cognition** | | | | | |
| MoCA | 27 | 2.18 | – | – | – |
| Digit Span Forward | 10.70 | 2.45 | 11.05 | 2.44 | -0.93(44) |
| Digit Span Backward | 9.33 | 2.37 | 9.32 | 6.65 | 0.02(44) |
| Digit Span Sequencing | 8.48 | 2.34 | 8.85 | 2.43 | -0.50(39) |
| Trails B-A (sec) | 36.49 | 23.20 | 31.67 | 25.65 | 0.67(44) |
| Stroop Inhibition Costs[2] | 0.77 | 0.24 | 0.81 | 0.19 | -0.52(45) |
| UFOV Processing Speed (ms) | 22.93 | 17.43 | 16.64 | 0.12 | 1.73(43) |
| UFOV Divided Attention (ms) | 46.99 | 25.84 | 23.27 | 14.36 | 3.83(43)*** |
| UFOV Selective Attention (ms) | 150.43 | 48.35 | 67.87 | 18.36 | 7.64(43)*** |
| **Vision** | | | | | |
| ETDRS Left Eye (logMAR) | 0.09 | 0.14 | -0.03 | 0.11 | 3.27(46)** |
| ETDRS Right Eye (logMAR) | 0.09 | 0.20 | -0.00 | 0.14 | 1.84(46) |

Independent samples t-tests were conducted on each outcome measure to compare performance between groups.

[1] Self-reported driving ability on a scale from 1–5, with 5 being excellent driving ability and 1 being very poor driving ability.

[2] Number of correct words uttered per second in the neutral condition from the incongruent condition.

PTA = Pure tone average a cross the 500 Hz, 1000 Hz, 2000 Hz, 3000 Hz, and 4000 Hz frequencies, higher value indicates a greater hearing loss. CDTT SRT = Canadian Digit Triplet Test Speech Reception Threshold, higher negative value indicates better performance. MoCA = Montreal Cognitive Assessment out of 30, higher value indicates better performance. Digit Span Forward, Backward, and Sequencing = higher value indicates better performance. Trails B-A = higher value indicates poorer performance. Stroop = higher value indicates poorer performance. ETDRS = Early Treatment Diabetic Retinopathy Study, lower value indicates better visual acuity. UFOV = Useful Field of View, higher value indicates poorer performance. *$p < 0.05$, **$p < 0.01$, ***$p < 0.001$.

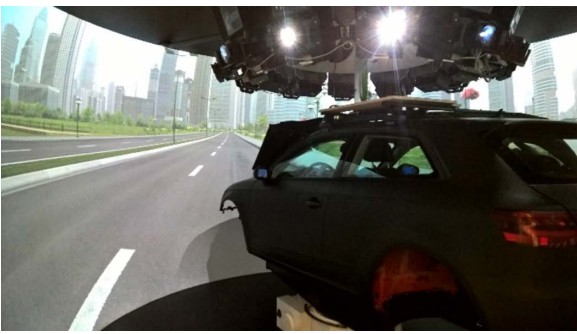

**Fig 2. DriverLab.**

## Apparatus

This study was conducted using a high-fidelity driving simulator, DriverLab, located at KITE, Toronto Rehabilitation Institute – University Health Network, Toronto, Canada (Fig 2). DriverLab is equipped with a full-sized passenger car (Audi A3) with all original internal components (e.g., steering wheel, gas/brake pedals, seats, dashboard). The vehicle is mounted on a turntable allowing 360 degrees of physical yaw rotation and is surrounded by a high-resolution visual projection system (Eyevis ESP-LWXT-2120, 1920 x 2000; 120 Hz), creating a seamless 360-degree field-of-view immersive experience. DriverLab is also equipped with a surround sound speaker system, which includes one subwoofer in the trunk of the vehicle and twelve speakers distributed around the vehicle's interior (Pioneer Elite VSX-45 Receiver, 5.1 sound; JL Audio powered sub; and Focal PC100, Focal PS130F, and Focal 100KRS speakers). Specifically, six loudspeakers are located in front of the participant (~24 inches in front of the driver's seat), four in the middle of the dashboard, one below the dashboard above the driver's feet, and one below the dashboard above the passenger's feet (~39 inches to the right of the driver's seat). Additionally, one loudspeaker is in the driver's side door (~12 inches in front of the driver's seat) and one loudspeaker in the front passenger side door (~12 inches in front and 48 inches to the right of the driver's seat). Four loudspeakers are also in the trunk of the vehicle (~68 inches behind the driver's seat) (See S2). The baseline decibel (dB) level when the vehicle is turned on and parked is 54 dB A, as measured using the REED R8050 Sound Level Meter (SLM). The SLM was calibrated using the 1/2" pre-polarised microphone (4230 pistonphone calibrator) to produce a 1000-Hz pure tone at 94 dB SPL in free field in the DriverLab environment. Driving scenarios were developed and presented using Scaner Studio software version 1.8.

## Experimental stimuli

**Listening task.** The Connected Speech Test (CST) was used as the listening task in this study. Participants listened to connected sentences spoken by a female talker who produced speech of average intelligibility for the average normal hearing younger adult in 6-talker babble [32,49]. In the CST, sentences are grouped or connected by a common discourse topic. Each topic consisted of 10 sentences, presented one at a time, with a six second inter-stimulus interval (ISI) between sentences. Participants were instructed to verbally repeat back the entire sentence exactly as they heard it and guess if unsure (see Data Analyses section below for information about scoring of the CST). The CST was presented through all DriverLab loudspeakers. The sound levels were measured using the REED R8050 SLM and were, on average, consistent across drives. Specifically, background noise, including the CST babble and DriverLab environmental and vehicle noises (e.g., engine noises, tires rumbling on the pavement, A/C fans) were measured three times, with an average of 64 dB A across all conditions. Listening difficulty was manipulated by adjusting the speech level of the target

female talker relative to the background noise to obtain two signal-to-noise ratios (SNR): an SNR of +4 dB (intended to be easier listening) and an SNR of 0 dB (intended to be harder listening). Sixteen topics in total were presented, eight in the high SNR conditions and eight in the low SNR conditions. We recognize that, while SNR manipulations were intended to create relatively easier and harder listening conditions, there may be age-related and individual differences in word recognition accuracy at these particular SNRs. Therefore, piloting of the experimental task, using the protocol described above with a small sample of younger ($N=6$) and older ($N=4$) adults, was conducted to ensure that the 0 dB SNR condition yielded poorer accuracy scores on the CST than the +4 dB SNR condition, in both age groups, without reaching floor or ceiling effects. Participants involved in the piloting phase did not participate in the main study.

**Driving task.** Participants completed three drives designed to mimic typical driving scenarios encountered in everyday life. During each drive, participants followed a lead car to control for speed, route, and total completion time across participants and to ensure alignment of driving along the route with the listening task trials/stimuli. Within each drive, driving difficulty (easier and harder) was manipulated both perceptually through visual clutter and cognitively/motorically through different driving tasks. Specifically, the easier driving difficulty section consisted of driving along a mostly straight, four-lane rural road (e.g., farm land, a few suburban houses, and little vehicular traffic) at 60 km/h with 1) a gradual curve, 2) two turns (one left-hand turn and one right-hand turn) at four-way intersections, and, 3) a stopped car in the left lane that had to be maneuvered around (Fig 3). The more difficult driving section involved driving along four- and six-lane city roads (e.g., many high-rise buildings, pedestrians on sidewalks, parked cars, and more vehicular traffic) at 60 km/h with 1) a gradual curve, 2) three turns (two left-hand turns and one right-hand turn) at four-way intersections, 3) a construction zone in the far-right lane, and, 4) a pedestrian crosswalk where no pedestrians actually crossed the road (Fig 3). Across participants, the order of the harder and easier driving sections within each drive was counterbalanced, but the section order was always the same for each driving scenario. See S3 Fig for a top-down view of an example driving scenario map.

## Procedure

Two sessions, each 1.5–2.0 hours in length, were completed.

Baseline and Practice Session: During the first session, all baseline assessments of sensory and cognitive functioning and questionnaires outlined above were administered. Following collection of the baseline assessment measures, participants completed a short practice while sitting in DriverLab with the car parked on the side of the road. During the practice, participants first demonstrated that they could correctly repeat one single sentence taken from the CST practice inventory, which was presented with no background babble. All participants correctly repeated at least 88% of the one sentence

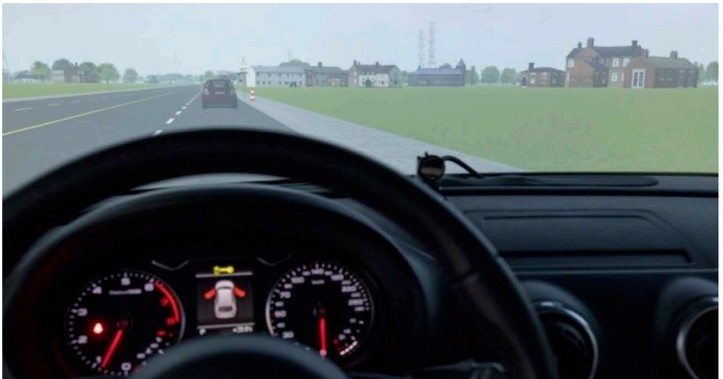 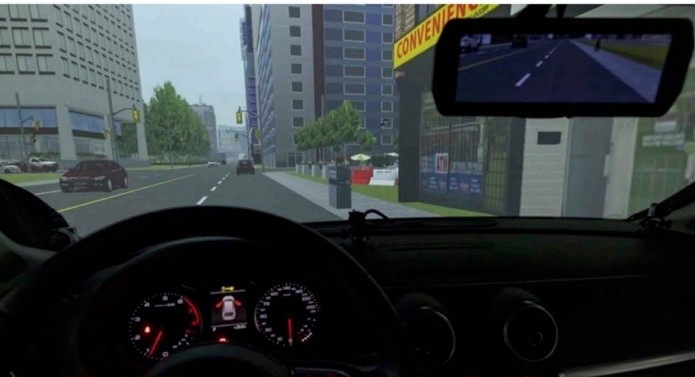

**Fig 3. Sample scenes from the Rural section of the drive (left) and the City section of the drive (right).**

presented in quiet, with all but 4 participants achieving 100% correct. Next, using the same CST practice inventory, participants practiced the task under both the + 4 dB SNR and 0 dB SNR listening conditions. Practice under both SNRs allowed participants to become familiar with the listening task and demonstrate that they could perform the experimental task with stimuli presented at the SNRs to be used in the experimental session.

Following the practice listening task, participants completed a 5-minute practice drive in DriverLab. The practice drive involved driving through a simulated environment with similar elements, but not identical, to the experimental drives. After the practice drive, participants rated their current feelings of motion sickness using the FMS [48]. The practice drive gave participants an opportunity to familiarize themselves with the vehicle controls of the driving simulator, practice following the lead car, and identify any symptoms of motion sickness that would preclude continued participation in the experimental session. Participants who proceeded to the experimental session had an average FMS score of 1.76/20 for younger adults and 1.58/20 for older adults after the practice drive.

Experimental Session: In the experimental session, participants completed a single-task listening condition (Listening Only), a single-task driving condition (Driving Only), and a dual-task driving-while-listening condition (Dual-Task; Fig 4). During the dual-task conditions, participants were asked to complete the two tasks at the same time. Specifically, participants were asked to drive safely and follow the lead car while also repeating back the sentences as best as they could. Participants were not instructed to prioritize either task. For both the single-task listening condition and dual-task condition, two different SNR conditions were completed (+4 dB SNR and 0 dB SNR). In total, there were five different experimental conditions. Each condition lasted approximately 8 minutes, with frequent breaks offered throughout the session. The FMS scale was also administered after each condition; the FMS rating averaged 1.35/20 for older and 1.42/20 for younger adults, with these low scores indicating that simulator sickness did not affect their performance. Experimental conditions were blocked by SNR, and their order was counterbalanced for both SNR and task type (single- versus dual-task). Participants completed five practice sentences at the start of the second session to remind them of the listening task, and five more practice sentences in the middle of the second session when the SNR changed [32].

*Listening only conditions:* Participants were seated in the driver's seat of the DriverLab vehicle while parked on the side of the road and repeated back each sentence from the CST. Participants were also asked to fixate on the road in front of

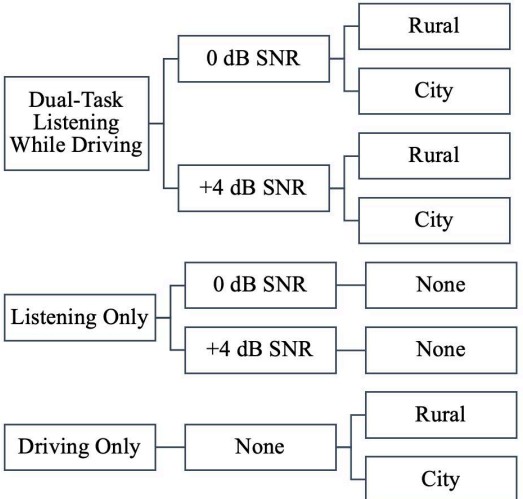

**Fig 4. Breakdown of the five conditions completed by all participants during the experimental session.**

them during the entire condition. A scene with a moderate number of visual stimuli (i.e., some high-rise buildings and some green space) was displayed to control the visuals across conditions. To ensure similarity of environmental background noise between Listening Only and Dual-Tasking conditions, the Listening Only conditions also contained environmental sounds that were present during the driving conditions. The easier listening condition included sentences (four topics) presented at +4 dB SNR, and the more difficult listening condition included sentences (four topics) presented at 0 dB SNR.

*Driving only conditions:* The Driving Only condition (no listening task) consisted of driving in the Rural and City scenarios with the elements described above. Participants were instructed to drive safely, follow the lead car, and refrain from talking during the entire drive.

*Dual-task listening while driving conditions:* In the Dual-Task listening while driving condition, participants completed the listening task while driving (+4 dB SNR and 0 dB SNR in separate drives).

## Data analyses

All statistical analyses were conducted with R statistical software [50], using alpha levels set to 0.05. Data met all necessary statistical assumptions for the analyses performed, including homogeneity of variance between groups, as assessed by the Levene's test for equality of variance ($p > 0.05$). Significant main and interaction effects were subjected to post-hoc analyses using Bonferroni correction and emmeans package [51]. Where necessary, violations of sphericity were accounted for using Greenhouse-Geisser corrections.

An a-priori power analysis was conducted using G*Power version 3.1.9.6 [(52] to determine the minimum sample size required to test study hypotheses. The a-priori power analysis was set to achieve 95% power for detecting a medium effect size of $f$ (0.25) on listening performance and driving performance, at a significance criterion of $\alpha = 0.05$. Results indicated the required sample size of $N = 18$ per group ($N = 36$ in total) for listening performance, and $N = 22$ per group ($N = 44$ in total) for driving performance, for a Repeated Measures Analysis of Variance (ANOVA), within-between interaction. The larger sample size (i.e., $N = 44$) was selected as the minimum for recruitment.

Raw Values – Listening Performance Analyses: The dependent variable for listening performance was word recognition accuracy measured as the percentage of 25 scoring keywords correctly repeated per topic with 4 topics per condition (i.e., 4 topics x 25 keywords per topic = 100 keywords scored and averaged per condition). A 2 x 2 x 3 mixed-factorial ANOVA was conducted with Age Group as a between-subjects factor (Younger Adults vs. Older Adults), and Listening Difficulty (+4 dB SNR vs. 0 dB SNR) and Task Type (Listening Only vs. Dual-Task Rural vs. Dual-Task City) as within-subject factors.

Raw Values – Driving Performance Analyses: The primary dependent variable for driving performance was the standard deviation of lane position (SDLP), defined as the dispersion of the lateral lane position [53], measured from the centre of the participant's vehicle to the edge of the immediate left lane line (i.e., white dashed lane line or yellow center line) in meters (m). A greater dispersion (higher values) indicates poorer driving performance. SDLP was selected as the key metric of interest because of its ability to measure variability throughout a drive across various road segments, its use as a common comparator measure in prior literature, and its demonstrated sensitivity to distraction-related effects on driving [16,17,20,54–56]. It has been common in prior studies to use speed-based metrics (e.g., mean and SD for speed), but they are less informative for the present study since participants were asked to follow a lead car. A 2 x 2 x 3 mixed-factorial ANOVA was conducted. Age Group was a between-subjects factor (Younger Adults vs. Older Adults), and Driving Difficulty (Rural vs. City) and Task Type (Driving Only vs. +4 dB SNR Dual-Task, vs. 0 dB SNR Dual-Task) were within-subject factors.

Proportional Dual-Task Cost Analyses: Dual-task costs were calculated as a proportional score which allowed for a more direct and meaningful comparison between listening and driving dual-task costs. Proportional dual-task costs were calculated using the following formulas:

(1)  Proportional Dual-Task Costs to Listening Accuracy: ((Single-Task – Dual-Task)/Single Task)

(2)  Proportional Dual-Task Costs to Driving SDLP: ((Dual-Task – Single-Task)/Single Task)

Positive values indicate performance reductions/dual-task costs, while negative values indicate performance enhancements/dual-task costs. Proportional dual-task costs were analyzed using a 2 x 2 x 2 mixed-factorial ANOVA separately for dual-task costs to listening performance and dual-task costs to driving performance, with proportional dual-task costs as the dependent variable. Age Group was a between-subjects factor (Younger Adults vs. Older Adults) and Listening Difficulty (0 dB SNR vs. + 4 dB SNR) and Driving Difficulty (Rural vs. City) were within-subject factors. Two participants were outliers (1 older adult for Listening Accuracy and 1 younger adult for SDLP), and their data was winsorized to within 3.5 standard deviations of the mean (no other outliers were identified for any other dependent measures). This transformation reduced the influence of outliers while maintaining sampling power that would otherwise have been compromised by full removal of the data. Using a 3.5 standard deviation cut-off is conservative to account for the heterogeneity typically observed in older adult populations and thereby limit the number of data points removed as outliers. Note, outliers were not attributable to technological or measurement errors. See S4 and S5 Files for further details regarding analyses and results.

One sample t-tests were also conducted to examine whether proportional dual-task effects significantly differed from zero. A value of zero would indicate neither a dual-task cost nor a dual-task benefit; therefore, any significant differences between proportional dual-task values and zero would indicate significant dual-task costs/benefits. See S4 and S5 Files for further details regarding analyses and results.

A series of Pearson's correlations were performed across all participants to examine whether listening and/or driving performance experimental measures were associated with hearing, and/or cognitive baseline measures. These correlations were intended to be exploratory and therefore no Bonferroni correction was used for multiple comparisons [57]. The extreme outlier data of one participant was excluded from two baseline measures (all UFOV sub-test scores and Trail Making). Four participants (3 older adults and 1 younger adult) had outlier data on some baseline measures and their data on those specific measures were winsorized to within 3.5 standard deviations of the mean. Since all CDTT SRT values were negative, values were changed to absolute values to allow for easier interpretation of the findings. To ensure consistency, PTA in the better ear, Trail Making scores, UFOV Processing Speed, Divided Attention, and Selective Attention sub-tests scores were reverse coded so that higher values indicated better performance across all baseline measures. Also see S6 File for correlations among sensory and cognitive measures.

The results section below will first present the data analyses for the listening performance variable of word recognition accuracy, followed by data analyses for the driving performance variable of SDLP, and concluding with a comparison between the influence of listening and driving performance on proportional dual-task costs. The listening and driving performance sections will start with presenting data analyses of the raw values in their original form to allow for direct examination of participant performance, followed by select proportional dual-task costs analyses to allow for examination of age-related differences in the magnitude of any proportional dual-task costs seen, and finally correlations between listening/driving performance raw values and hearing and cognitive measures.

## Results

### Listening performance – Accuracy

There was a significant main effect of Listening Difficulty: for both older and younger adults, there were higher accuracy scores in the + 4 dB SNR condition compared to the 0 dB SNR condition, $F(1, 46) = 496.92$, $p < 0.001$, $\eta^2 = .58$ (Fig 5). There was also a significant main effect of Task Type, $F(1.76, 81.10) = 9.03$, $p < 0.001$, $\eta^2 = .02$, and a two-way interaction effect of Task Type and Listening Difficulty ($p = 0.009$), which was qualified by a three-way Listening Difficulty x Task Type x Age Group interaction effect $F(1.84, 84.66) = 3.94$, $p = 0.026$, $\eta^2 = .01$ (Fig 5). For younger adults, there were no significant differences between Single- and Dual-Task conditions in either 0 dB SNR or +4 dB SNR Listening Conditions (all comparisons were $p > 0.05$) (Fig 5). In contrast, older adults' word recognition accuracy depended on Task Type (generally poorer listening performance with increasing task difficulty; Dual-Task City < Dual-Task Rural < Single Task Listening) and on Listening Difficulty. Specifically, post-hoc analyses revealed that, in the 0 dB SNR Listening Condition, older adults had

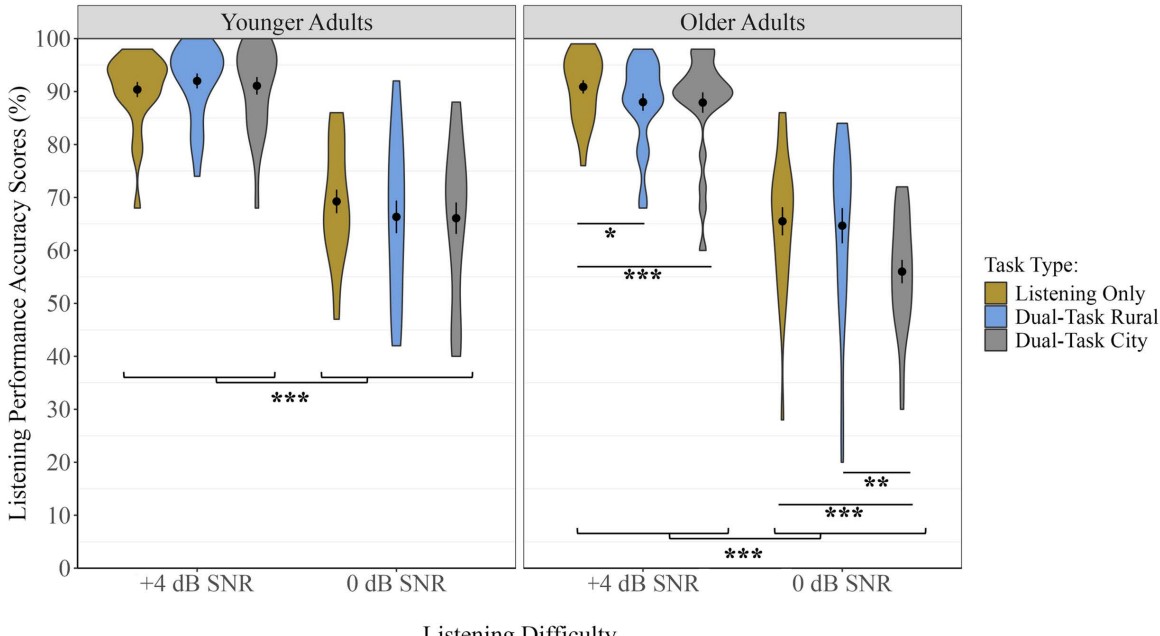

**Fig 5. Listening task performance percent correct word recognition accuracy scores for each Age Group, Listening Difficulty, and Task Type condition.** Each violin plot represents the frequency of the data at each point on the y-axis. The center dot represents the mean. Error bars represent ±1 SE. * $p < 0.05$, ** $p < 0.01$, *** $p < 0.001$.

higher accuracy scores in the Listening-only compared to the Dual-Task City condition, $t(46) = -5.18$, $p < 0.001$ and higher accuracy scores in the Dual-Task Rural compared to the Dual-Task City condition, $t(46) = -3.67$, $p = 0.002$, but there were no significant differences between Listening-only compared to the Dual-Task Rural condition, $t(46) = 0.37$, $p = 1.00$. In the ±4 dB SNR Listening Condition, older adults had higher accuracy scores in the Listening-only compared to the Dual-Task City condition, $t(46) = -2.58$, $p = 0.040$, higher accuracy scores in the Listening-only compared to the Dual-Task Rural condition, $t(46) = 2.98$, $p = 0.014$, but there were no significant differences in the Dual-Task City condition compared to the Dual-Task Rural condition, $t(46) = -0.06$, $p = 1.00$.

Proportional dual-task costs to listening performance were consistent with those presented in the raw values (see S4 for full analyses and figures), except for unique information regarding whether differences observed between dual- and single-task conditions are significantly greater for one group compared to the other. Specifically, these analyses found that older adults had greater proportional dual-task costs to listening performance compared to younger adults in the City section under the +4 dB SNR Listening Condition, $t(46) = 2.20$, $p = 0.033$ and the 0 dB SNR Listening Condition, $t(46) = 2.30$, $p = 0.026$, and in the Rural section under the +4 dB SNR Listening Condition, $t(46) = 3.20$, $p = 0.003$, but not the 0 dB SNR Listening Condition, $t(46) = -0.35$, $p = 0.727$ (S4).

### Correlations: (Listening accuracy)

A series of Pearson's correlations were conducted across all participants to assess the linear associations between CST word recognition accuracy scores across conditions and the baseline measures of hearing (PTA in the better ear, CDTT SRT) and cognition (MoCA, Trail Making, Digit Span, Stroop Inhibition Cost, UFOV Processing Speed, UFOV Divided Attention, and UFOV Selective Attention) across all participants (Fig 6).

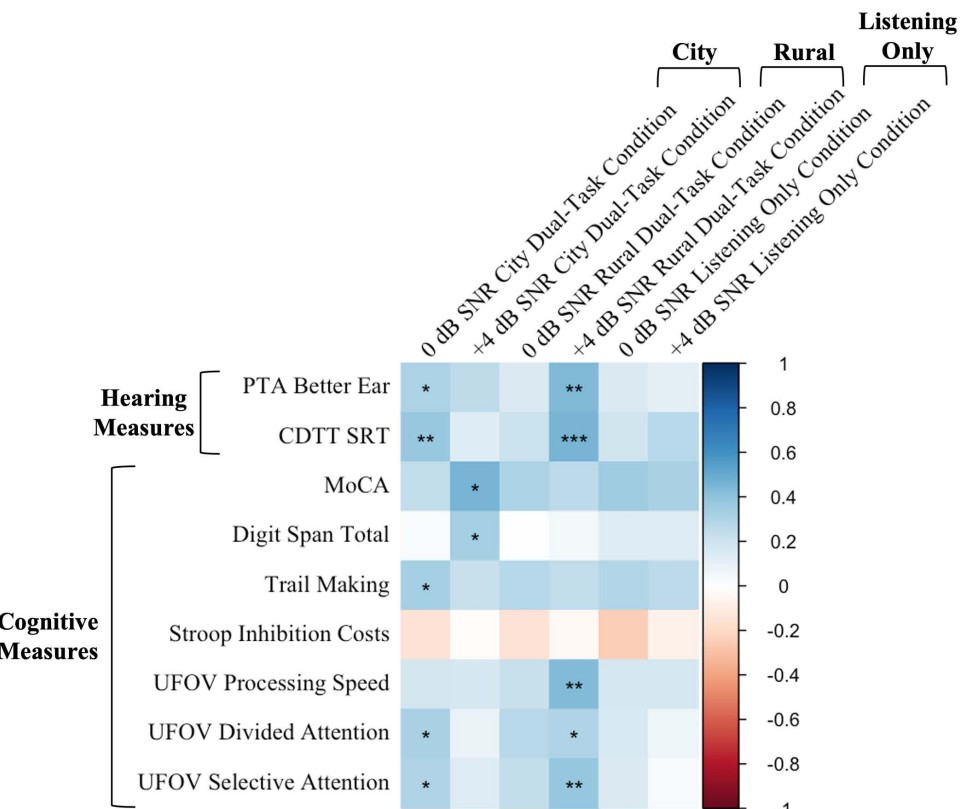

**Fig 6. Correlations between word recognition accuracy scores in the six different experimental conditions (columns) and baseline measures (rows).** To ensure consistency, PTA in the better ear, Trail Making scores, UFOV Processing Speed, Divided Attention, and Selective Attention sub-tests scores were reverse coded so that higher values indicated better performance across all measures. A positive r value indicates a positive relationship (blue) and a negative r value indicates a negative relationship (red). *$p < 0.05$, **$p < 0.01$, ***$p < 0.001$.

**Hearing**: There was a positive correlation between PTA and word recognition accuracy scores in the 0 dB SNR City Dual-Task condition, $r(45) = 0.32$, $p = 0.031$ and in the +4 dB SNR Rural Dual-Task condition, $r(45) = 0.44$, $p = 0.002$, with better hearing being associated with better listening accuracy. There was a positive correlation between CDTT SRTs and word recognition accuracy scores in the 0 dB SNR City Dual-Task condition, $r(46) = 0.38$, $p = 0.008$ and in the +4 dB SNR Rural Dual-Task condition, $r(46) = 0.46$, $p < 0.001$, demonstrating that better speech-in-noise thresholds were associated with better word recognition accuracy.

**Cognition:** There was a positive correlation between MoCA scores and word recognition accuracy in the +4 dB SNR City Dual-Task condition, $r(22) = 0.46$, $p = 0.023$, demonstrating that better cognition was associated with better word recognition accuracy. There was also a positive correlation between Digit Span scores and word recognition accuracy in the +4 dB SNR City Dual-Task condition, $r(39) = 0.33$, $p = 0.033$, demonstrating that better working memory was associated with better word recognition accuracy. There was also a positive correlation between Trail Making scores and word recognition accuracy in the 0 dB SNR City Dual-Task condition, $r(44) = 0.33$, $p = 0.025$. There were no significant correlations between Stroop scores and word recognition accuracy for any of the experimental conditions ($p > 0.05$).

There was a positive correlation between UFOV Processing Speed and word recognition accuracy in the +4 dB SNR Rural Dual-Task condition, $r(43) = 0.43$, $p = 0.003$, demonstrating that faster visual processing speed was associated with better word recognition accuracy. There was also a positive correlation between UFOV Divided Attention and word

recognition accuracy in the 0 dB SNR City Dual-Task condition, $r(43) = 0.33$, $p = 0.030$, and the +4 dB SNR Rural Dual-Task condition, $r(43) = 0.31$, $p = 0.042$, demonstrating that better visual divided attention was associated with better word recognition accuracy. There was also a positive correlation between UFOV Selective Attention and word recognition accuracy in the 0 dB SNR City Dual-Task condition, $r(43) = 0.31$, $p = 0.039$, and in the +4 dB SNR Rural Dual-Task condition, $r(43) = 0.38$, $p = 0.010$, demonstrating that better visual selective attention was associated with better word recognition accuracy.

### Driving performance – Standard deviation of lane position (SDLP)

With regards to SDLP, significant main effects of Age Group ($F(1, 46) = 10.40$, $p = 0.002$, $\eta^2 = .12$), Driving Difficulty ($F(1, 46) = 295.03$, $p < 0.001$, $\eta^2 = .36$), and Task Type ($F(2, 91.98) = 10.50$, $p < 0.001$, $\eta^2 = .05$) were observed, as well as significant two-way interaction effects between Age Group x Driving Difficulty and Driving Difficulty x Task Type ($p < 0.05$). These effects were qualified by a three-way Age Group x Driving Difficulty x Task Type interaction effect, $F(1.98, 91.23) = 9.19$, $p < 0.001$, $\eta^2 = .02$ (Fig 7). Specifically, significant effects of Task Type were only observed in the more difficult City section and not the Rural section. In the Rural sections, neither older nor younger adults had significantly different SDLP among conditions (all comparisons were $p > 0.05$) (Fig 7). In the City section, older adults demonstrated increasingly greater SDLP (poorer driving performance) with increasing task difficulty levels (Single < +4-Dual < 0-Dual). Specifically, post-hoc comparisons revealed that, compared to the Driving Only condition, older adults in the City section had significantly higher SDLP in the 0 dB SNR Dual-Task condition, $t(46) = 8.48$, $p < 0.001$, and in the +4 dB SNR Dual-Task condition, $t(46) = 3.70$ $p = 0.002$. Furthermore, SDLP was significantly higher in the 0 dB SNR Dual-Task condition compared to the +4 dB SNR Dual-Task condition, $t(46) = 4.69$, $p < 0.001$. In contrast, younger adults demonstrated greater SDLP (poorer driving performance) in the Driving Only condition compared to the 0 dB SNR Dual-Task condition, $t(46) = 4.39$, $p < 0.001$ and in

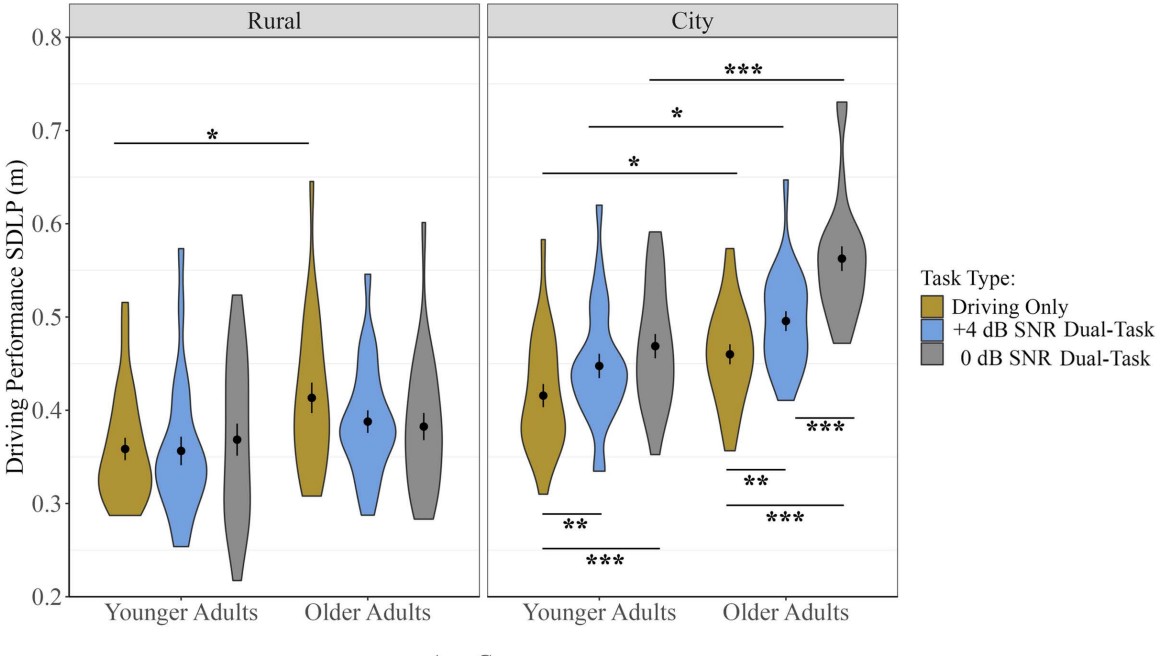

**Fig 7. SDLP in meters, for each Age Group, Task Type, and Driving Difficulty condition.** Each violin plot represents the frequency of the data at each point on the y-axis. The center dot represents the mean. Error bars represent ±1 SE. *$p < 0.05$, **$p < 0.01$, ***$p < 0.001$.

the +4 dB SNR Dual-Task condition, $t(46)$ = 3.31, $p$ = 0.006 (Single-Task < +4 dB SNR Dual-Task and Single-Task < 0 dB SNR Dual-Task); however, there was no difference in SDLP for the 0 dB SNR Dual-Task condition compared to the +4 dB SNR Dual-Task condition, $t(46)$ = 1.49, $p$ = 0.431.

When comparing driving performance across participant groups, in the Rural section, older adults had higher SDLP in the Driving Only condition compared to younger adults, $t(46)$ = 2.70, $p$ = 0.010, but there were no age-related differences observed in the other two conditions ($p$ > 0.05) (Fig 7). In contrast, older adults had higher SDLP in all conditions in the City section compared to younger adults. Specifically, post-hoc comparisons revealed that, in the City section, compared to younger adults, older adults had significantly higher SDLP in the Driving Only condition, $t(46)$ = 2.70, $p$ = 0.010, in the +4 dB SNR Dual-Task condition, $t(46)$ = 2.85, $p$ = 0.007; and in the 0 dB SNR Dual-Task condition, $t(46)$ = 5.01, $p$ < 0.001.

Proportional dual-task costs to driving performance were consistent with those presented in the raw values (see S5 for full analyses and figures), except for unique information regarding whether the differences observed between dual- and single-task conditions are significantly greater for one group compared to the other. Specifically, in the City section, under the 0 dB SNR Listening Difficulty condition, older adults had significantly higher proportional dual-task costs to SDLP compared to younger adults, $t(46)$ = 2.19, $p$ = 0.033; however, proportional dual-task costs to SDLP were not significantly different between younger and older adults in any other combination of conditions ($p$ > 0.05) (S5).

### Correlations: (Standard deviation of lane position)

A series of Pearson's correlations were conducted across all participants to assess the linear associations between driving performance (SDLP) across conditions and the baseline measures of hearing (PTA in the better ear, CDTT SRT) and cognition (MoCA, Trail Making, Digit Span, Stroop Inhibition Cost, UFOV Processing Speed, UFOV Divided Attention, and UFOV Selective Attention) (Fig 8).

**Hearing:** There was a negative correlation between CDTT SRTs and SDLP in four of the six conditions (0 dB SNR City Dual-Task condition, $r(46)$ = -0.37, $p$ = 0.011; +4 dB SNR City Dual-Task condition, $r(46)$ = -0.39, $p$ = 0.006; 0 dB SNR Rural Dual-Task condition, $r(46)$ = -0.30, $p$ = 0.039 Rural Driving Only condition, $r(46)$ = -0.39, $p$ = 0.006), demonstrating that poorer hearing was associated with poorer driving (i.e., higher SDLP values). There were no significant correlations between PTA and driving performance in any of the experimental conditions ($p$ > 0.05).

**Cognition:** There were no significant correlations between the cognitive measures of MoCA, Digit Span, Trail Making, Stroop, UFOV Processing Speed, and UFOV Divided Attention and driving performance in any of the experimental conditions ($p$ > 0.05). There was a negative correlation between UFOV Selective Attention and SDLP in the 0 dB SNR City Dual-Task condition, $r(43)$ = -0.42, $p$ = 0.004, in the City Driving Only condition, $r(43)$ = -0.33, $p$ = 0.029, and in the Rural Driving Only condition, $r(43)$ = -0.30, $p$ = 0.049, demonstrating that poorer visual selective attention was associated with poorer driving.

### Relative costs to listening and driving performance: Comparison between listening and driving performance proportional dual-task costs

To determine whether participants had greater dual-task costs to listening or driving performance, comparisons between proportional dual-task costs to listening performance and proportional dual-task costs to driving performance were conducted separately for younger and older adults using paired sample t-tests. Comparisons are only reported for proportional dual-task costs in the City section since the primary analyses reported above showed the greatest effects in these driving conditions. See S7 File for Rural section analyses.

In the City section, results for the older group showed significantly greater proportional dual-task costs to SDLP compared to word recognition accuracy in the 0 dB SNR Listening Difficulty condition, $t(23)$ = -2.76, $p$ = 0.011. In contrast, for younger adults, the effects only approached significance for the same condition $t(23)$ = -2.03, $p$ = 0.054, and should be interpreted with caution. In the +4 dB SNR Listening Difficulty condition, younger adults showed significantly greater

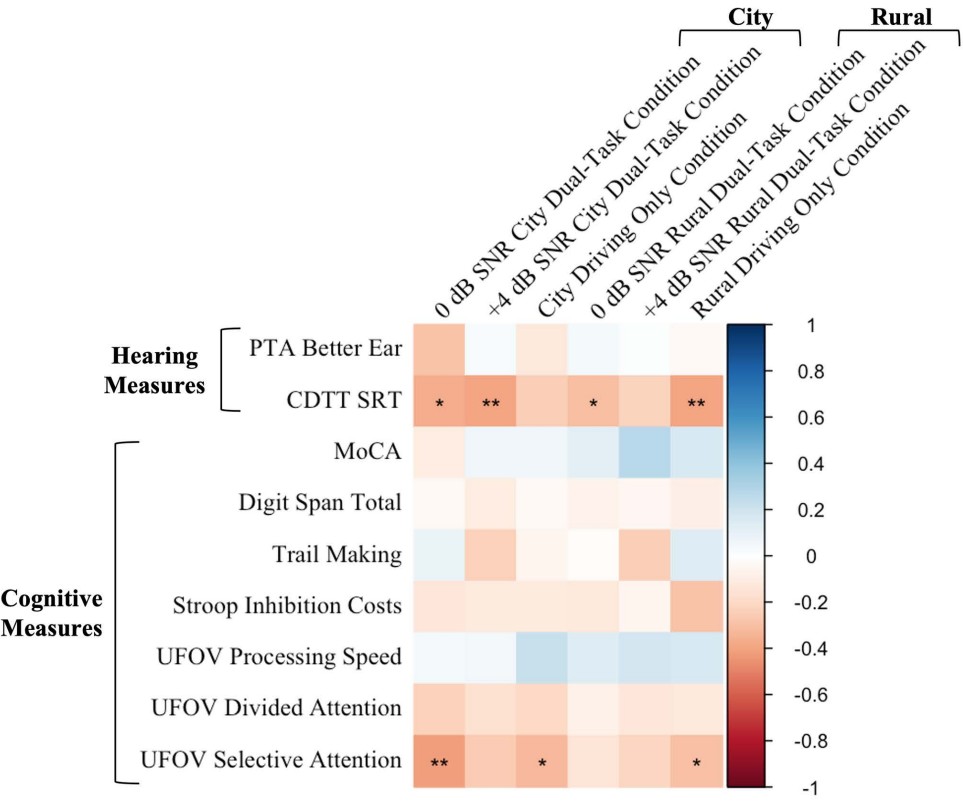

**Fig 8. Correlations between driving performance SDLP in the six different experimental conditions (columns) and baseline measures (rows).** To ensure consistency, PTA in the better ear, Trail Making scores, UFOV Processing Speed, Divided Attention, and Selective Attention sub-tests scores were reverse coded so that higher values indicated better performance across all baseline measures. Lower values for SDLP indicate better performance. A positive r value indicates a positive relationship (blue) and a negative r value indicates a negative relationship (red). *$p < 0.05$, **$p < 0.01$, ***$p < 0.001$.

proportional dual-task costs to SDLP compared to listening accuracy, $t(23) = -3.94$, $p < 0.001$. However, for older adults, effects only approached significance when the proportional dual-task costs to SDLP were compared to proportional dual-task costs to listening accuracy for the same condition, $t(23) = -2.00$, $p = 0.057$, and should be interpreted with caution (Fig 9).

## Discussion

### Summary

This study examined the effects of listening while driving in younger and older adults with clinically normal hearing thresholds, visual acuity, and cognition using a high-fidelity driving simulator designed to introduce multisensory, multitasking challenges. In terms of word recognition performance, when having to manage the tasks of listening and driving at the same time (compared to listening while parked), younger adults showed no performance decrements under any conditions, whereas older adults demonstrated decrements in word recognition accuracy, particularly during the most demanding conditions (i.e., City, 0 dB SNR). Age-related differences in dual-task costs to word recognition accuracy may be attributable to the higher speech-in-noise thresholds and elevated high frequency pure-tone thresholds for older adults compared to younger adults (CDTT SRT and better ear PTA respectively, Table 1).

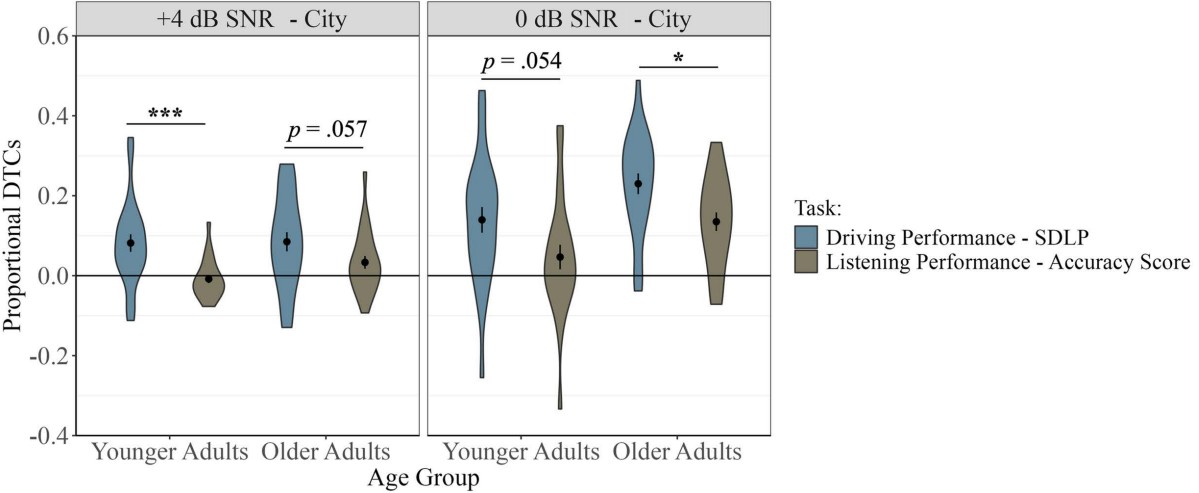

**Fig 9. Comparison of proportional dual-task costs (DTCs) between driving performance (SDLP) and listening performance (accuracy scores), for each Age Group, in the City section in +4 dB SNR and 0 dB SNR Listening Difficulty conditions.** Each violin plot represents the frequency of the data at each point on the y-axis. The center dot represents the mean. Error bars represent ±1 SE. Positive values indicate a dual-task cost (poorer performance in Dual compared to Single Task). $*p < 0.05$, $***p < 0.001$.

In terms of driving performance, when having to manage the tasks of driving and listening concurrently (compared to driving only), both older and younger adults showed decrements to their driving performance (more variable lane keeping), which were most pronounced under the most challenging conditions (City, 0 dB SNR). However, the magnitude of these dual-task costs under challenging conditions was greater in older adults than in younger adults. These observed patterns of dual-task costs to driving under the most challenging conditions may be attributable, in part, to age-related differences, with poorer speech-in-noise thresholds (CDTT SRT scores), visual acuity, and visual selective attention in older compared to younger adults (Table 1).

### Effects of age, driving difficulty, and listening difficulty on dual-task *word recognition accuracy*

The current study provided novel insights toward understanding how listening abilities are affected during driving and how these effects differ in older and younger adults with clinically normal hearing, vision, and cognition. With the use of more simplistic tasks in most previous dual-task studies of listening effort, it was not clear how earlier findings might generalize to more ecologically valid tasks, and how age-related differences may influence cognitive resource allocation under more realistic conditions. The current study found that younger adults' word recognition accuracy was similar across conditions and was not affected by driving difficulty or dual-tasking. However, for older adults, word recognition accuracy was poorer in the more difficult driving condition (i.e., City) compared to the listening only condition, regardless of listening difficulty (i.e., +4 dB SNR and 0 dB SNR). Further, older adults' word recognition accuracy was also poorer during the easier driving condition (i.e., Rural) compared to the listening only condition when listening was less difficult.

The current study found that poorer speech-in-noise thresholds (i.e., CDTT SRT scores) were associated with poorer word recognition accuracy in almost all dual-task conditions. There were also some associations between word recognition accuracy and poorer performance on other sensory and cognitive measures (e.g., PTA, MoCA, Trail Making, Digit Span, UFOV) under some dual-task conditions, but no associations were observed for any single-task conditions (see Fig 6). Participants with better sensory and cognitive abilities may have experienced less demand on cognitive resources, resulting in better listening performance under dual-task conditions. It is important to acknowledge that these associations

are exploratory in nature, and no corrections for multiple comparisons were applied. Consequently, these findings should be interpreted with caution.

The effects of age on dual-task costs to word recognition accuracy cannot be directly compared to previous studies of auditory distraction while driving because those studies lacked a single auditory task condition, precluding the single-task vs. dual-task comparison needed to calculate dual-task costs. We can compare the current effects of dual-tasking on word recognition accuracy to typical dual-task listening effort studies conducted in controlled, isolated environments using simple and artificial secondary tasks (e.g., visual tracking task, tactile pattern analysis) to help better understand whether similar findings may be observed during more complex and realistic tasks. Previous studies examining the effects of age on listening difficulty have found overall greater listening effort as a consequence of dual-tasking in older adults compared to younger adults, evidenced by either similar dual-task costs between age groups or no dual-task costs for both age groups on the primary listening task performance, but greater dual-task costs to secondary task performance in older adults compared to younger adults [8–10,58]. The current findings expand on this work by demonstrating that only older adults exhibited dual-task costs to both word recognition performance and driving performance under the easier (+4 dB SNR) and harder (0 dB SNR) listening conditions and harder driving conditions (City). Younger adults, however, only exhibit dual-task costs to driving performance when the driving demands were high (City). Therefore, it appears that dual-task costs increase as the listening and concurrently performed tasks become more complex, requiring increased use of cognitive resources, for which older and younger adults may allocate their resources differently.

### Effects of age, driving difficulty, and listening difficulty on dual-task *driving performance*

The current study provided novel insights that increase our understanding of how word recognition in easier (+4 dB SNR) and harder (0 dB SNR) conditions affect driving performance in younger and older adults. Integrating a speech-in-noise task that strategically targeted supra-threshold sensory and cognitive abilities known to decline with older age allowed us to study dual-task costs in realistic situations. The testing paradigm used in the current study provides a more ecologically valid approach than traditional lab-based studies for investigating performance by reflecting common listening while driving scenarios in everyday life. Overall, older adults had poorer driving performance (greater lane variability) compared to younger adults, consistent with previous studies and with our predictions [16,17,20,54]. Results also showed that both younger and older adults had larger lane deviations in dual- compared to single- task conditions, but only in the most difficult City driving conditions (not in Rural driving). For older adults, these dual-task costs were particularly pronounced when the word recognition task was more difficult, whereas the driving performance of younger adults was not affected by easier or harder word recognition conditions. Increased listening difficulty may have resulted in more resources being allocated to listening, thereby reducing the cognitive capacity available for driving.

The relationship between driving performance and baseline measures of hearing and cognition help to better understand which specific domains of cognitive and sensory functioning may be most important in managing the competing demands of listening while driving. For example, better speech-in-noise thresholds (i.e., CDTT SRT) were associated with better driving performance across almost all driving conditions. Notably, audiometric thresholds (i.e., PTAs) were not associated with driving performance under any condition. PTAs may not be a reliable indicator of speech understanding abilities in noisy environments, as older adults with clinically normal audiometric pure-tone thresholds often still perform more poorly on speech-in-noise tests than younger adults with normal hearing [59,60]. The current findings appear to extend prior findings to more complex driving tasks, whereby better speech-in-noise thresholds rather than audibility (lower PTA) are associated with better management of the relative demands of performing a speech-in-noise word recognition task while driving. Further, better visual selective attention was associated with better driving performance, particularly under more demanding driving conditions with high visual clutter (i.e., City), and were generally not associated with driving performance under less difficult driving and visually sparse conditions (i.e., Rural). Perhaps better visual abilities allow for better processing of dynamic, visually complex scenes, allowing for better management of the demands of navigating

complex environments. Interestingly, there was no relationship between driving performance and other cognitive measures such as working memory, inhibition, and executive functioning; these cognitive measures were also not significantly different between age groups. Taken together, it appears that speech-in-noise thresholds and visual selective attention may have been particularly relevant to the listening while driving task in the current study. It is important to note that strict eligibility criteria with respect to PTA and cognition limited variability on these measures, restricting the ability to find any relationships with these measures to driving performance; furthermore, as these associations are exploratory and no corrections for multiple comparisons were applied, the findings should be interpreted with caution.

The observed associations between driving task performance and sensory/cognitive measures may also help explain several inconsistent findings in previous driving literature, whereby some studies have reported age-related differences in driving performance under conditions of auditory distraction [17,20,56], while others reported no age-related differences [16,18,19,54,55]. One potential reason for these discrepancies could be due to the type of driving tasks used in previous studies that are often not clearly described, making it difficult to categorize driving task difficulty levels. However, studies that do provide clear descriptions consistently report no age-related differences in dual-task costs to driving performance, regardless of whether the driving scenarios were relatively simple (e.g., driving straight down a two-lane road with minimal amounts of buildings and traffic; [16,19,55,56]) or relatively complex (e.g., driving through intersections with many buildings and high levels of traffic; [18,54]). Another potential reason for these discrepancies could be due to the type of auditory tasks used in previous studies. Specifically, most previous dual-task driving studies have implemented an auditory secondary task to introduce overall cognitive demands rather than strategically selecting an auditory task that may be sensitive to known age-related changes in hearing, such as speech-in-noise tasks. Another contributing factor may be differences between studies in the availability of supportive linguistic context, including lexical, syntactic, and semantic cues within and between sentences. Older adults often perform similarly, if not better than younger adults when such cues are present [61], with their linguistic expertise potentially reducing cognitive demands compared to younger adults. The more simplistic auditory tasks used in previous studies (e.g., repeating digits, auditory Go-No-Go task) are also similar to the baseline measures of working memory (i.e., Digit Span) and inhibitory control (i.e., Stroop) used in the current study, which did not differ between age groups and also showed no correlations with driving performance within this healthy younger and older adult sample. However, it is difficult to make clear interpretations on the effects of dual-tasking on driving performance in younger and older adults in previous studies because the complexities of both the driving task and the listening task need to be considered. Previous driving studies using complex driving tasks have not measured auditory task performance, limiting interpretations of whether performing a concurrent listening task while driving causes one to exceed their spare capacity limit, resulting in disengagement from the listening task all together (i.e., poorer listening task performance) to maintain driving performance. Taken together, driving and listening tasks may be most sensitive to age-related differences in dual-task costs when engagement on both tasks is still high and cognitive capacity is not exceeded. Tasks that are too perceptually/cognitively simple may produce minimal performance differences, while overly complex tasks may lead to task disengagement, both limiting the ability to detect group differences.

### Relative task costs: Differences in relative dual-task effects on listening vs. driving performance

Evaluating performance on the complex, multisensory, and potentially safety-critical task of driving while listening provides unique insights toward understanding how each of these tasks influence one another and whether this relative influence is affected by older age. The current study found that both older and younger adults had greater proportional dual-task costs to driving compared to listening. Specifically, for older adults, this pattern was only observed under the most difficult listening and driving conditions. For younger adults, greater dual-task costs to driving compared to listening were only observed under conditions when listening was easier (+4 dB SNR) and driving was harder (City), for which there were dual-task costs to driving performance but dual-task benefits to listening performance (i.e., benefits meaning better performance on the listening task under dual-task conditions compared to single-task conditions). It is important to note that greater costs

do not necessarily translate into compromised driving safety in this study, as measurements of safety were not directly examined, and measured lane position variability generally remained within safe limits. Participants averaged fewer than two unintended lane departures per 8-minute condition, with no significant differences between conditions or groups. This low number of lane departures suggests that driving behaviour was generally safe and the dual-task was manageable. However, the dual-task costs to driving overall may indicate a general *increased risk* of compromised safety.

Previous dual-task studies using other safety critical tasks, such as standing or walking, often report that older adults typically prioritize balance/gait (i.e., the more safety critical task), evidenced by similar gait/balance performance in dual- and single-task conditions, but poorer listening performance under dual- compared to single-task conditions [13–15]. However, it has also been shown that postural control can become poorer under dual-task conditions when listening conditions are highly adverse [62]. Therefore, the extent to which dual-task costs are observed is likely due to the relative difficulty of the tasks being concurrently performed. Specifically, when both tasks are too easy there may be no dual-task costs, as observed for the younger adults in the current study. Tasks that are moderately difficult demonstrate the largest dual-task costs on one or both tasks, evidenced in the current study by dual-task costs to both tasks under difficult driving conditions (City) and both listening difficulties for older adults, and only dual-task costs to driving under difficult driving conditions (City) for younger adults.

## Limitations and future directions

First, due to the strict eligibility criteria of the current study with respect to sensory, cognitive, and motor function, the sample of older adults was mostly highly educated, healthy, physically, and socially active individuals. The range of our baseline measures of sensory/cognitive abilities was restricted to the upper range of abilities and thus, our older adult sample is not representative of the broader, heterogenous older adult population. Therefore, the age-related differences in listening and driving performance evidenced here may be conservative relative to what would be observed in the general population, where greater age-related effects may be predicted.

The older adults in this study had significantly more years of driving experience ($M = 49.05$ years) than younger adults ($M = 9.88$ years). Age-related differences in driving experience between age groups in the current study may have reduced the age-related effects on some of our outcome measures, particularly under the easier driving conditions (Rural), because older adults could compensate for potential age-related declines in sensory/cognitive/motor abilities by relying on well-practiced driving skills and expertise.

Third, while the CST simulates realistic speech-in-noise challenges and allows for standardized listening responses across participants and conditions, it does not realistically mimic natural conversations. Thus, participants may have been able to repeat the sentences correctly, without deeper comprehension of the information. This shallower depth of processing required by the CST task may not have been as cognitively or emotionally demanding as natural conversations, in which deeper comprehension is often required (e.g., information to be remembered in future). Future studies could explore the effects of depth of processing on listening and driving performance by incorporating auditory memory tasks to see how much information participants remembered and/or use a diverse set of listening tasks requiring different types of responses and varying degrees of processing.

## Conclusion

The goal of this study was to examine age-related differences in performance on the realistic and complex task of listening while driving. The findings provide important new insights into how increases in driving and listening difficulties negatively affect dual-task performance, which may be due to age-related differences in sensory and cognitive abilities even when these abilities are within the normal range. The results demonstrated that only older adults showed significantly poorer word recognition accuracy in the dual-task condition compared to the single-task listening condition across both SNR conditions. Age-related differences in word recognition accuracy, particularly under dual-task conditions, was associated with poorer

speech-in-noise thresholds. Further, both older and younger adults showed significantly poorer driving performance in the dual-task condition compared to the single-task driving conditions, but only in the more difficult driving condition (i.e., City). Dual-task costs to driving under these most challenging conditions were associated with speech-in-noise thresholds and visual selective attention. Future research could examine ways to potentially mitigate these dual-task effects by, for example, working memory training to help reduce the cognitive load associated with speech understanding during driving, optimization of the vehicle acoustics or use of technology to improve the SNR, or minimization of auditory distractions, especially when driving situations become more complex (e.g., in the City). More generally, these strategies may enhance communication and comfort, increase independence and social participation, and reduce driving-related risks.

## Supporting information

**S1 Table. Total *N*'s for each demographic and assessment outcome measure.**
(DOCX)

**S2 Fig. Side-view and top-down view of the speaker configuration in DriverLab.**
(DOCX)

**S3 Fig. Top-down view of a scenario map.**
(DOCX)

**S4 File. Listening performance – Proportional dual-task costs (accuracy) and Determining whether proportional dual-task costs differ from zero.**
(DOCX)

**S5 File. Driving performance – Proportional dual-task costs (standard deviation of lane position) and Determining whether proportional dual-task costs differ from zero.**
(DOCX)

**S6 File. Associations among sensory and cognitive baseline measures.**
(DOCX)

**S7 File. Relative task costs between listening and driving performance in the rural sections.**
(DOCX)

## Acknowledgments

Thank you to Colin Stoddart, Dr. Bruce Haycock, Susan Gorski, Roger Montgomery, and Robert Shewaga for their technical assistance, to Joseph Rovetti and Kay Wright-Whyte for their assistance with methodological considerations, and to Yadurshana Sivashankar, Lianna Montanari, and Joanne Nuque for their assistance in data collection.

Photo credit: Tim Fraser.

## Author contributions

**Conceptualization:** Katherine Bak, M. Kathleen Pichora-Fuller, Frank A Russo, Jennifer L Campos.

**Data curation:** Katherine Bak, Kristen Arnold, Lena Darakjian, Jennifer L Campos.

**Formal analysis:** Katherine Bak.

**Funding acquisition:** Jennifer L Campos.

**Investigation:** Katherine Bak, Kristen Arnold, Lena Darakjian.

**Methodology:** Katherine Bak, M. Kathleen Pichora-Fuller, Frank A Russo, Jennifer L Campos.

**Project administration:** Katherine Bak, Kristen Arnold, Lena Darakjian, Jennifer L Campos.

**Resources:** Katherine Bak, Jennifer L Campos.

**Supervision:** Katherine Bak, Jennifer L Campos.

**Visualization:** Katherine Bak.

**Writing – original draft:** Katherine Bak, Jennifer L Campos.

**Writing – review & editing:** Katherine Bak, Kristen Arnold, Lena Darakjian, M. Kathleen Pichora-Fuller, Frank A Russo, Jennifer L Campos.

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
