## [Decision Letter · Decision Letter 0]

18 Nov 2024

PONE-D-24-41749Dual-task costs of listening while driving in older and younger adultsPLOS ONE

Dear Dr. Bak,

Thank you for submitting your manuscript to PLOS ONE. After careful consideration, we feel that it has merit but does not fully meet PLOS ONE’s publication criteria as it currently stands. Therefore, we invite you to submit a revised version of the manuscript that addresses the points raised during the review process.

We look forward to receiving your revised manuscript.

Kind regards,

Sangamanatha Ankmnal Veeranna, Ph.D.

Academic Editor

PLOS ONE

Additional Editor Comments:

Did any of the participants were multilingual? If yes, do the authors think that it could

Lines 164-169: Did the authors screen hearing or did obtain hearing thresholds? Hearing screening is conducted usually at one intensity level (usually 25 dB HL).

Why was hearing not tested at 6 and 8 kHz? Hearing loss at these frequencies is very common in older adults. This could affect speech recognition in difficult listening conditions.

I would plot hearing thresholds for each test frequency for both groups.

Lines 169-171: 15 dB interaural difference was recommended by the American Academy of Otorhinolaryngology- Head and Neck Surgery (https://www.entnet.org/resource/position-statement-red-flags-warning-of-ear-disease/). Suen et al (2021) reported the prevalence of asymmetrical hearing loss for different criteria (10-, 15- and 20-dB HL).

Line 171: PTA means pure tone average or pure tone thresholds.

Lines 172-175: The authors should provide more details about the digit triplet test. How was it administered? Was this test online?

There are no sufficient details about the calibration. How did the authors measure the sound levels in the DriverLab environment? How far were the speakers from the listener?

Lines 251-252: How was the sound level was measured? Did sound levels vary based on different driving scenarios? Depending on the scenario, should the background noise levels vary?

Line 259: Regarding the connected speech test (CST), why didn’t the author use the latest version of the CST (Saleh et al., 2020)? The older version of the CST has a higher noise floor (Saleh et al., 2020) and the accent differed. How will the authors justify the noise floor in the CST recording and its effect on the results of the current study?

Provide information about the scoring for the CST.

Lines 273-275: Provide more information about the piloting.

Lines 357-363: Isn’t it common practice to use 0.5 as the effect size for calculation? On what basis were the effect sizes chosen?

Isn’t it that the signal-to-noise ratio expression is usually in dB?

For all the ANOVA analyses, the authors should report the homogeneity of variance results.

Lines 397-399: Any reference or justification for using 3.5 SD?

Lines 410-411: Exploratory analyses are not used for testing hypotheses. The authors should adjust the p-value for multiple comparisons.

Lines 411-413: How did the authors decide that the data is an outlier?

The result section is very difficult to follow and lacks clarity.

It will be easier for the readers to follow if both results and discussion are in the same order.

Saleh HK, Folkeard P, Macpherson E, Scollie S. Adaptation of the Connected Speech Test: Rerecording and Passage Equivalency. Am J Audiol. 2020 Jun 8;29(2):259-264. doi: 10.1044/2019_AJA-19-00052. Epub 2020 Mar 20. PMID: 32196353.

Reviewers' comments:

Reviewer's Responses to Questions

**Comments to the Author**

1. Is the manuscript technically sound, and do the data support the conclusions?

Reviewer #1: Partly

2. Has the statistical analysis been performed appropriately and rigorously? 

Reviewer #1: Yes

3. Have the authors made all data underlying the findings in their manuscript fully available?

Reviewer #1: No

4. Is the manuscript presented in an intelligible fashion and written in standard English?

Reviewer #1: Yes

5. Review Comments to the Author

Reviewer #1: In this manuscript, the authors examined the effect of performing a challenging listening task on driving performance using a dual-task paradigm in groups of older and younger adults. The aim was to compare the experience of listening effort between the two groups. Overall, I found the manuscript to be of good quality, particularly in terms of the novelty of the research question and the design. However, I have several significant concerns that need to be addressed before I can recommend the manuscript for publication.

My main concern is the lack of clarity in presenting the study's primary outcome, which suggests no significant difference in listening effort between the groups as significant deterioration in secondary task performance was observed in both groups. This key finding was not clearly articulated in the manuscript, which I believe could mislead readers. The authors predominantly focused on the deterioration in primary task performance, which is not, by itself, a definitive measure of increased listening effort. Below, I provide more specific comments and suggestions for improvement.

1- It would be helpful if the authors referred to the listening task as the “primary task” and the driving task as the “secondary task,” as these are the commonly used terms in dual-task paradigms. Using this terminology would improve clarity and make the manuscript easier to follow.

2- 2- The results section is overly detailed and difficult to follow, even with the use of subheadings. I suggest rewriting it more concisely, focusing on the key outcomes. Additionally, some of the results reported were not clearly linked to the study's stated aims. For instance, the differences in primary task performance across different SNR conditions were not mentioned as objectives. Another example is the association between performance and sensory/cognitive measures, which should be clearly justified as part of the study’s goals. Please revise the section to ensure that all reported results align with the study’s aims.

3- The authors compared dual-task performance using two different methods: proportional and non-proportional, which may cause confusion for the reader. It would be more effective to choose a single method for assessing changes in dual-task performance and clearly state this in the analysis. This would avoid the need to report results twice for both the primary and secondary tasks which would make the findings easier to follow.

4- “Raw values – Driving performance (SDLP)”; I do not understand how SDLP is an acronym to this statement.

5- Section starting line 533; I do not understand the significance of this analyses and how is it different from the post-hoc analysis performed in the earlier sections.

6- Lines 645-648; Typically, dual-task cost is calculated based on the deterioration of secondary task performance. This statement is somewhat misleading, especially given the significant deterioration observed in secondary task performance in both groups.

7- Lines 170-171: what is the degree of hearing loss in the worse hearing ear on this basis for these two participants? Safer to remove these two participants to avoid the risk of the inclusion of participants with unilateral hearing loss.

8- I suggest the authors arrange the discussion following the same order used in the results section. E.g. association between performance in the dual task and different measure of speech in noise cognition etc. is reported at the end of the results section but discussed early in the discussion.

9- When discussing the association between performance on the primary and secondary tasks with measures such as speech-in-noise, cognition, and vision (e.g., in the paragraph starting on line 718), please clarify whether the results pertain to the older group, the younger group, or both.

Minor comments

1- Mines 111-112; worth adding a comment of how this represents real life situation where sometimes roads can be noisy making listening difficult.

2- Line 152; so they are not all natives? how frequently was English used in there life then? although some can learn English at young age and speak it fluently, but that does not make it as strong as there mother language.

3- Lines 405-406; please provide a reference.

4- Lines 742 to 747, I do not understand the argument that the authors are trying to make here. Are they referring to their own work? Please clarify.

5- Lines 759-762, again, how is this related to the previous paragraph? Are the authors referring to their own study? Please clarify?

6- Sentence in line 792 contradicts the rest of the paragraph.

6. PLOS authors have the option to publish the peer review history of their article (what does this mean? ). If published, this will include your full peer review and any attached files.

**Do you want your identity to be public for this peer review?** For information about this choice, including consent withdrawal, please see our Privacy Policy .

Reviewer #1: **Yes: ** Sara Alhanbali

---

## [Author Response · Author response to Decision Letter 1]

6 Dec 2024

We would like to sincerely thank the editor and reviewers for taking the time to provide such thoughtful comments towards improving our manuscript. This document lists all comments specific to each reviewer, interleaved with the authors’ responses and, when applicable, references to the changes made to the manuscript. We are confident that these changes have improved the manuscript and we hope that you will again consider it for publication in PLOS ONE.

1. Responses to Journal Requirements:

1.1. Please ensure that your manuscript meets PLOS ONE's style requirements, including those for file naming. The PLOS ONE style templates can be found at

Response: Thank you for the prompt. We have double checked that the manuscript complies with PLOS ONE’s style requirements.

1.2. We note that you have indicated that there are restrictions to data sharing for this study. PLOS only allows data to be available upon request if there are legal or ethical restrictions on sharing data publicly. For more information on unacceptable data access restrictions, please see http://journals.plos.org/plosone/s/data-availability#loc-unacceptable-data-access-restrictions. If there are ethical or legal restrictions on sharing a de-identified data set, please explain them in detail (e.g., data contain potentially identifying or sensitive patient information, data are owned by a third-party organization, etc.) and who has imposed them (e.g., a Research Ethics Committee or Institutional Review Board, etc.). Please also provide contact information for a data access committee, ethics committee, or other institutional body to which data requests may be sent.

Response: Thank you for the additional information regarding data sharing restrictions. We have obtained a formal letter from our Research Ethics Board which explains in detail the ethical restrictions on sharing a de-identified dataset. Please see the document titled “UHN REB Letter on Sharing De-Identified Data” in the PLOS ONE portal. For your reference, the body of the letter states, “I am writing in my capacity as the Board Lead for the Research Ethics Board (REB) at the University Health Network and REB coordinator of the REB file regarding the manuscript titled “Dual-task costs of listening while driving in older and younger adults,” authored by Katherine Bak, Kristen Arnold, Lena Darakijan, M. Kathleen Pichora-Fuller, Frank A. Russo, and Jennifer L. Campos. The protocol and consent form for this study, as approved by our REB, specifically restrict the sharing of de-identified data publicly. This restriction is based on the language included in the consent process, which explicitly states that data collected from participants will only be accessible to and shared within the study team. Consequently, sharing the de-identified dataset with external entities, such as other researchers or organizations, would constitute a breach of the conditions under which participant consent was obtained. The only exception to this is if sharing of this data is required by law. Even if sharing were possible, the data that UHN REB would allow would be extremely limited due to the nature of the information collected for the study. Such REB restrictions would be in place to minimize any potential risks of re-identification or misinterpretation outside of its intended context. UHN REB recognizes that such limitations would inherently impact the reproducibility of any analyses derived from the dataset. These restrictions were imposed by The University Health Network's Research Ethics Board, an independent body responsible for ensuring compliance with ethical standards and participant protections. Any requests for further information or clarifications about the study's data-sharing policies may be directed to our REB office.”

Contact Information for the REB:

Research Ethics Board

University Health Network

700 University Ave. 4th floor

Toronto, ON, M5G 1Z5

Email: reb@uhn.ca

Phone: 416-581-7849 (main line)

Thank you for your understanding of the ethical considerations that underpin our decision-making in this matter. If you require further clarification, please do not hesitate to contact me or the REB directly.

Sincerely,

Lorraine Pirrie

1.3. Please review your reference list to ensure that it is complete and correct. If you have cited papers that have been retracted, please include the rationale for doing so in the manuscript text, or remove these references and replace them with relevant current references. Any changes to the reference list should be mentioned in the rebuttal letter that accompanies your revised manuscript. If you need to cite a retracted article, indicate the article’s retracted status in the References list and also include a citation and full reference for the retraction notice.

Response: Thank you for the prompt. The reference list has been reviewed and is complete and correct. URLs were added where relevant.

Note that we removed the Suen et al (2021) reference for asymmetrical hearing loss criteria, as the Editor kindly pointed out that the main purpose of this paper is to report on the prevalence of asymmetrical hearing loss for different criteria. As such, we have added the correct reference to the reference list (American Academy of Otorhinolaryngology- Head and Neck Surgery [AAO-HNS]. Position Statement: Red Flags-Warning of Ear Disease, 2021).

We also added the Rothman 1990 reference to support the rationale for not correcting for multiple comparisons in our correlations plots (“Rothman K. J. (1990). No adjustments are needed for multiple comparisons. Epidemiology (Cambridge, Mass.), 1(1), 43–46.”).

2. Responses to Editor Comments:

2.1. Did any of the participants were multilingual? If yes, do the authors think that it could

Response: Thank you for this insightful comment. We did not specifically ask for multilingualism; however, we did ask for participants’ primary language of choice. All participants who responded to this question stated that English was their primary language of choice, except for two participants who stated Persian (1 younger adult) and Farsi (1 older adult) as their primary language of choice. We have added these additional details to the manuscript (Page 5 lines 156-158) which now reads “All participants provided written informed consent, spoke fluent English, learned English by the age of, 5 and reported English as their primary language of choice (with the exception of one younger adult and one older adult participant)…” Please also see section 3, responses to Reviewer #1, Comment 3.11.

2.2. Lines 164-169: Did the authors screen hearing or did obtain hearing thresholds? Hearing screening is conducted usually at one intensity level (usually 25 dB HL).

Response: We did not do a hearing screening at one intensity level. Instead, we screened hearing based on audiometric pure-tone air-conduction thresholds obtained using ShoeBoxTM (see the detailed procedures on the pure-tone audiometry on Page 6 lines 171-174). We have updated the manuscript to further clarify what kind of hearing test was completed. Page 6 lines 171-174 now reads “A screening hearing test was administered in-person using ShoeboxTM (Version 5.6.7); participants’ pure-tone audiometric air-conduction thresholds (dB HL) were measured in each ear across five frequencies of 500 Hz, 1000 Hz, 2000 Hz, 3000 Hz, and 4000 Hz to determine eligibility.” We describe this as a screening because we obtained pure-tone thresholds for the purposes of our research, but we did not conduct full audiometry for diagnostic purposes (e.g., testing did not include bone-condition or masked thresholds).

2.3. Why was hearing not tested at 6 and 8 kHz? Hearing loss at these frequencies is very common in older adults. This could affect speech recognition in difficult listening conditions.

Response: Thank you for this comment. Our ShoeBoxTM hearing test did measure pure-tone audiometric thresholds (dB HL) in each ear across the frequencies of 6000 Hz and 8000 Hz, in addition to the 500 Hz, 1000 Hz, 2000 Hz, 3000 Hz, and 4000 Hz thresholds used for eligibility criteria. We used the frequencies of 500 Hz, 1000 Hz, 2000 Hz, 3000 Hz, and 4000 Hz as these frequencies are used to classify individuals into normal hearing and hearing loss categories, as per the World Health Organization (WHO) criteria. We have added a plot of hearing thresholds across the frequencies of 500 Hz to 8000 Hz for each group. Please also see our response to comment 2.4 below.

2.4. I would plot hearing thresholds for each test frequency for both groups.

Response: Thank you for the helpful suggestion to include a plot of hearing thresholds for each test frequency for both groups. We have added an additional figure of hearing thresholds at each frequency (500 Hz – 8000 Hz) for both groups, which will be inserted on Page 7 of the Methods section, at the end of the Hearing subsection (please also see below for new figure).

Fig 1. Hearing thresholds at each frequency (500 Hz – 8000 Hz) and ear for both younger and older adult groups. The circle and triangle symbols represent the means. Error bars represent ±SE.

2.5. Lines 169-171: 15-dB interaural difference was recommended by the American Academy of Otorhinolaryngology- Head and Neck Surgery (https://www.entnet.org/resource/position-statement-red-flags-warning-of-ear-disease/). Suen et al (2021) reported the prevalence of asymmetrical hearing loss for different criteria (10-, 15- and 20-dB HL).

Response: Thank you for pointing out the correct citation. We have updated the reference to include the American Academy of Otolaryngology-Head and Neck Surgery (AAO-HNS) online statement which defines asymmetrical hearing loss using the criteria outlined in the manuscript (Page 6, line 178).

2.6. Line 171: PTA means pure tone average or pure tone thresholds.

Response: Thank you for pointing out that we did not clearly state what PTA stands for. We have added “pure-tone threshold average” before the first mention of PTA in the manuscript (Page 6, line 178).

2.7. Lines 172-175: The authors should provide more details about the digit triplet test. How was it administered? Was this test online?

Response: Thank you for pointing out that we did not clearly state what the digit triplet test is measuring and whether it was tested in-person or online. We have updated the manuscript to state that the CDTT was conducted in-person and that it is a speech-in-noise test that measures speech reception thresholds in noise (Page 6, line 181-182). We also note that the digit triplet test was administered as per the procedures outlined in reference 33.

2.8. There are no sufficient details about the calibration. How did the authors measure the sound levels in the DriverLab environment? How far were the speakers from the listener?

Response: The REED R8050 Sound Level Meter (SLM) was used to measure the baseline noise level (when the car was turned on and parked) and the average background noise level (including the CST babble and DriverLab environmental and vehicle noises (e.g., engine noises, tires rumbling on the pavement, A/C fans)) at head level when seated (Page 10 line 270-271, and Page 11 lines 305 – 307). The SLM was first calibrated using the 1⁄2” pre-polarised microphone (4230 pistonphone calibrator) to produce a 1000-Hz pure tone at 94 dB SPL in free field while sitting in DriverLab. These details regarding calibration have been added to the manuscript on Pages 10 lines 271- 272 and Page 11 line 288 - 289. Please also see response to comment 2.9 below.

The distance of the loudspeakers and the speaker configuration has also been updated in the manuscript on Page 10, lines 262 to 270. Additionally, a diagram of the speaker locations has been included in the supplementary materials (please also see below for new figure). In total there were twelve loudspeakers. Specifically, six loudspeakers are located in front of the participant (~24 inches in front of the driver’s seat), four in the middle of the dashboard, one below the dashboard above the driver’s feet, and one below the dashboard above the passenger’s feet (~39 inches to the right of the driver’s seat). Additionally, one loudspeaker is in the driver’s side door (~12 inches in front of the driver’s seat) and one loudspeaker in the front passenger side door (~12 inches in front and 48 inches to the right of the driver’s seat). Four loudspeakers are also in the trunk of the vehicle (~68 inches behind the driver’s seat) (See S2).

S2 Figure. Side-view and top-down view of the speaker configuration in DriverLab.

2.9. Lines 251-252: How was the sound level was measured? Did sound levels vary based on different driving scenarios? Depending on the scenario, should the background noise levels vary?

Response: As stated in our response to comment 2.8 above, sound level measurements were obtained using the REED R8050 SLM (which was calibrated before each use) from the position of the head when seated. The average baseline noise level and background noise level were measured, and the CST signal was presented at a +4 dB SNR and a 0 dB SNR as stated in the manuscript. Each driving scenario contained the same driving elements and tasks (e.g., same curves, same number of turns at the same angle, etc.), but in a different order. As such, sound levels were on average consistent across drives. We would also like to note that the background driving environmental noise levels (e.g., variations due to acceleration, speed) may have varied across a scenario and/or between participants depending on how the vehicle was maneuvered. For example, one participant may have accelerated more quickly during a straightaway compared to another participant who may have accelerated more slowly, causing an increase in acceleration-related noise levels. However, we are confident that any potential differences in environmental noise levels across participants and scenarios were minimal as, 1) participants were required to follow a lead car which limited fluctuations in speed throughout the scenarios, and 2) participants never had to bring the car to a full stop, as yield signs were used instead of stop signs at intersections, which limited large fluctuations in acceleration throughout the scenarios. In addition to adding information about obtaining sound measurements using the REED R8050 SLM, we have also added that sound levels were on average consistent across drives (Page 11 lines 303-304).

2.10. Line 259: Regarding the connected speech test (CST), why didn’t the author use the latest version of the CST (Saleh et al., 2020)? The older version of the CST has a higher noise floor (Saleh et al., 2020) and the accent differed. How will the authors justify the noise floor in the CST recording and its effect on the results of the current study? Saleh HK, Folkeard P, Macpherson E, Scollie S. Adaptation of the Connected Speech Test: Rerecording and Passage Equivalency. Am J Audiol. 2020 Jun 8;29(2):259-264. doi: 10.1044/2019_AJA-19-00052. Epub 2020 Mar 20. PMID: 32196353.

Response: Thank you for bringing this important paper to our attention. Methodological considerations for this study were finalized in 2019, before publication of this latest version of the CST. We also conducted a follow-up study with older adults with and without age-related hearing loss, which uses the same methodology described in this manuscript. The original CST was adapted by grouping topics to ensure equivalency, addressing differences observed in individuals with hearing loss when using the original topic set. To our knowledge, the latest version of the CST has not yet been tested in those with hearing loss. Therefore, to maintain consistency across our two studies, we chose to use the original CST with topic equivalency for both individuals with normal hearing and hearing loss. Additionally, the latest CST development paper did not compare intelligibility scores between the original test materials and their new test materials. As such, it is unclear whether the differences in accent and

---

## [Decision Letter · Decision Letter 1]

19 Jan 2025

PONE-D-24-41749R1Dual-task costs of listening while driving in older and younger adultsPLOS ONE

Dear Dr. Bak,

Thank you for submitting your manuscript to PLOS ONE. After careful consideration, we feel that it has merit but does not fully meet PLOS ONE’s publication criteria as it currently stands. Therefore, we invite you to submit a revised version of the manuscript that addresses the points raised during the review process.

I agree with reviewer 1 that the result section is too lengthy and confusing. The authors should highlight the key findings they want readers to focus on, ensuring that the most important messages are clear and well-emphasized. Additionally, please make sure to clearly address the calibration process and any questions related to hearing thresholds.

We look forward to receiving your revised manuscript.

Kind regards,

Sangamanatha Ankmnal Veeranna, Ph.D.

Academic Editor

PLOS ONE

Additional Editor Comments:

Lines 144-145: The authors have introduced these two lines without a proper context in the introduction. Reviewer-1 has raised similar concerns.

Hearing testing: Based on what the authors have mentioned, it is not a screening test, the authors have conducted an air-conduction hearing test.

Figure 1 clearly illustrates that hearing thresholds in older adults are elevated compared to those of younger adults. However, it is important to note that bone conduction thresholds are not included in the audiogram. This raises the question: Could the older participants have slight sensorineural hearing loss? According to the WHO (1991) guidelines on the grades of hearing impairment, individuals with thresholds between 15-20 dB may experience hearing difficulties. Figure 1 indicates that some individuals in the older group had elevated hearing thresholds at 3 and 4 kHz, which could have contributed to their reduced performance on these tasks. Are we underestimating the impact of these elevated thresholds on complex dual-task performance? Hearing thresholds at high frequencies play a crucial role in recognizing speech in challenging listening environments. Consider adding these thresholds as a covariate in the analyses to assess if they have any impact.

Additionally, the authors referenced the WHO (1991) guidelines for calculating hearing thresholds at 0.5, 1, 2, 3, and 4 kHz frequencies. However, the table in the guidelines specifically recommends calculating hearing thresholds only at 0.5, 1, and 2 kHz. The authors have to clarify this.

The authors do not clearly state the calibration procedure which is an important part of this manuscript if someone wants to replicate it.

Lines 284-285: What sound pressure levels were measured? To determine the sound pressure level, what type of sound was used? Was it the target sentence or the background noise?

How many times did these calibrations were conducted?

I think it is appropriate to report SNR in dB throughout the manuscript, consistently.

In the previous revision, I asked about the piloting process. How was it conducted, and how many participants were involved? Additionally, how many participants participated in both the pilot and the main study? If some participants were involved in both, they would have been familiar with the study. Do the authors believe that this familiarity with the study and the tasks could have influenced the results?

The authors have responded to my question, stating that "it is not necessary to apply a p-value correction for multiple comparisons when the data under evaluation are not random numbers but actual observations of nature" (Rothman K. J., 1990, Epidemiology). However, I find their response regarding p-value adjustments for multiple comparisons unconvincing. If the authors do not support adjusting the p-value, they should refrain from using it altogether in this manuscript. The authors have applied a Bonferroni correction in the manuscript (lines 378-379), but this is not the only available method. The False Discovery Rate (FDR) is another appropriate approach the authors could consider.

The results section remains difficult to follow. The authors should focus on presenting the key findings that they want the readers to take away, ensuring clarity and emphasis on the most important messages.

Reviewers' comments:

Reviewer's Responses to Questions

**Comments to the Author**

1. If the authors have adequately addressed your comments raised in a previous round of review and you feel that this manuscript is now acceptable for publication, you may indicate that here to bypass the “Comments to the Author” section, enter your conflict of interest statement in the “Confidential to Editor” section, and submit your "Accept" recommendation.

Reviewer #1: (No Response)

2. Is the manuscript technically sound, and do the data support the conclusions?

Reviewer #1: Partly

3. Has the statistical analysis been performed appropriately and rigorously? 

Reviewer #1: No

4. Have the authors made all data underlying the findings in their manuscript fully available?

Reviewer #1: Yes

5. Is the manuscript presented in an intelligible fashion and written in standard English?

Reviewer #1: Yes

6. Review Comments to the Author

Reviewer #1: 1- I still find this manuscript lengthy and challenging to follow. The extensive analyses, lengthy results section, and inclusion of 11 figures make it a difficult read. I strongly suggest that the authors consider streamlining the analyses by focusing on a single method that effectively conveys the main message of the manuscript. This would improve clarity and readability. Additionally, I am not entirely convinced that proportional analyses serve as an appropriate measure of prioritization in the context of the task at hand.

2- “Thank you for bringing this to our attention. Although we recognize that traditionally tasks are often referred to as primary and secondary, this is typically under situations when participants are explicitly asked to prioritize one task over the other (or where this is otherwise presumed). We intentionally did not specifically ask participants to prioritize one task over the other or introduce one task as the primary task and one task as the secondary task in our participant instructions. This was intentional as we wanted the paradigm to be more comparable to real-world scenarios for which tasks are not always prioritized in such a way, and prioritization may change rapidly depending on task demands and individual differences. As well, in a real-world listening while driving scenario, we would not typically expect participants to think of the primary task as the listening task and the secondary task as the driving task, due to the safety critical nature of driving. As such, we wanted to see how and whether participants would prioritize one task over the other. We have added a sentence to the Methods section to clarify that participants were not asked to prioritize one task over the other, which states, “During the dual-task conditions, participants were not instructed to prioritize either task.” (Page 14, lines 370 - 371).”

Based on this, I am not entirely confident that the term "listening effort" accurately describes what is being assessed in this context. It is difficult to imagine a scenario where the quality of driving is compromised in order to prioritize listening. Since driving quality pertains to a safety-critical issue, it is highly unlikely that listening would ever be treated as the primary task. As I previously mentioned, in the literature on listening effort, listening is typically regarded as the primary task that consumes the majority of cognitive resources, with any other task being considered secondary. This framework applies even when participants are explicitly instructed to prioritize either the primary or secondary task. Therefore, I believe the author needs to provide a stronger justification for why the measured construct should be considered "listening effort" rather than, for example, "driving effort."

3- “We have also added a sentence to include our secondary goal of exploring the associations among primary outcome measures and sensory and cognitive abilities. Specifically, “A secondary goal of the current study was to explore associations among listening and driving performance and measures of hearing, vision, and cognition.” (Page 5, lines 145-146).”

Inserting this to the end of the introduction in this way comes out of nowhere as it his hard for reader to understand the motive for this investigation given that related literature was not addressed in the introduction.

4- Lines 144-145; it is good that the authors have added this as a goal of the study, however, the rationale for exploring these association and the implications of doing so should have been introduced to the reader somewhere in the introduction of the manuscript.

5- Line 155; please remove the word “and” from the end of the sentence “and reported”.

6- Line 177; add what you mentioned in the rebuttal letter here about these two participants.

7- Line 209; mention what they were used for, i.e. some further analysis to address the research questions later? Same applies to wherever the same statement was mentioned in the methods section.

8- Line 347: Not clear how? What were the instructions exactly?

9- “The current study found that both older and younger adults had greater proportional dual-task costs to driving compared to listening. Specifically, for older adults, this pattern of prioritization was only observed under the most difficult listening and driving conditions.”

Need to explain how proportional analysis addresses the concept of task prioritization more explicitly.

10- For younger adults, greater dual-task costs to driving compared to listening were only observed under conditions when listening was easier (+4 dB SNR) and driving was harder (City), for which there were dual-task costs to driving performance but dual-task benefits to listening performance”

Why is this explained as prioritization? Could it not simply be that one task is easier than the other? In dual-task paradigms, participants often tend to prioritize the easier task. In this particular case, this might be especially true given that it involves a simulated driving scenario, where participants are aware that there is no real risk of harm.

11- 893-894: what do you mean by contextual speech cues?

7. PLOS authors have the option to publish the peer review history of their article (what does this mean? ). If published, this will include your full peer review and any attached files.

**Do you want your identity to be public for this peer review?** For information about this choice, including consent withdrawal, please see our Privacy Policy .

Reviewer #1: **Yes: ** Sara Alhanbali

---

## [Author Response · Author response to Decision Letter 2]

28 Feb 2025

We would like to sincerely thank the editor and reviewers for taking the time to provide additional thoughtful comments towards improving our manuscript. This document lists all comments specific to each reviewer, interleaved with the authors’ responses and, when applicable, references to the changes made to the manuscript. We are confident that these changes have improved the manuscript and we hope that you will again consider it for publication in PLOS ONE.

1. Responses to Editor Comments:

1.1. Lines 144-145, “A secondary goal of the current study was to explore associations among listening and driving performance and measures of hearing, vision, and cognition”: The authors have introduced these two lines without a proper context in the introduction. Reviewer-1 has raised similar concerns.

Response: Thank you for pointing this out. We have added some additional context to the introduction of the manuscript to justify the inclusion of our secondary objectives, which can be found on Pages 3 and 4 lines 113 to 118 and states: “Previous literature has shown associations between hearing and cognition in healthy older adults (Lin, 2011, 2013). However, to our knowledge few experimental studies have examined how hearing measures (e.g., pure-tone threshold average [PTA], words-in-noise test scores) and cognitive measures (e.g., working memory, executive control) are independently associated with performance on more complex listening-while-driving dual-tasks.”

We also removed vision measures from the secondary objective, as the main motivation for this secondary goal comes from previous literature examining the link between hearing and cognition specifically. While we acknowledge that driving is a highly visual task, and that basic vision tests might be associated with dual-tasking performance, our only measure of vision was visual acuity. Since participants typically correct for reduced visual acuity with prescription glasses, the range of visual abilities that could be assessed was limited. Notably, visual acuity showed few to no significant associations with our primary outcome measures. Therefore, vision has been excluded from the correlation analyses and related figures.

1.2. Hearing testing: Based on what the authors have mentioned, it is not a screening test, the authors have conducted an air-conduction hearing test. Figure 1 clearly illustrates that hearing thresholds in older adults are elevated compared to those of younger adults. However, it is important to note that bone conduction thresholds are not included in the audiogram. This raises the question: Could the older participants have slight sensorineural hearing loss? According to the WHO (1991) guidelines on the grades of hearing impairment, individuals with thresholds between 15-20 dB may experience hearing difficulties. Figure 1 indicates that some individuals in the older group had elevated hearing thresholds at 3 and 4 kHz, which could have contributed to their reduced performance on these tasks. Are we underestimating the impact of these elevated thresholds on complex dual-task performance? Hearing thresholds at high frequencies play a crucial role in recognizing speech in challenging listening environments. Consider adding these thresholds as a covariate in the analyses to assess if they have any impact.

Response: Thank you for your insightful comment. We fully agree that elevated high-frequency pure-tone thresholds may play a role in speech understanding in challenging listening environments. Our goal, however, was not to minimize the influence of these age-related changes, but rather to note their relevance and importance as a motivation for this study (e.g., Page 1 lines 48-53, Page 3 lines 108-113, and Page 4 lines 128-132). It is important to note that this study specifically excluded older adults with pure-tone thresholds exceeding clinically significant levels (average of ≥25 dB HL in the better ear across 500, 1000, 2000, 3000, and 4000 Hz, the main speech frequencies). We intentionally included older adults with the commonly observed elevated thresholds (relative to younger adults). This was intentional to be able to examine how age-related changes in hearing abilities affect dual-task performance. Indeed, one of the aims of conducting correlations between primary outcome measures (e.g., driving performance) and baseline sensory measures, such as pure-tone thresholds, was to investigate the potential associations.

Upon further review, we identified a section in the Discussion that did not adequately address the potential influence of pure-tone thresholds on listening performance. We have now clarified this in the Discussion Summary (Page 27 lines 857–860), which states: “Age-related differences in dual-task costs to word recognition accuracy may be attributable to the higher speech-in-noise thresholds and elevated high-frequency pure-tone thresholds for older adults compared to younger adults (CDTT SRT and PTA Better Ear respectively, Table 1).”

1.3. Additionally, the authors referenced the WHO (1991) guidelines for calculating hearing thresholds at 0.5, 1, 2, 3, and 4 kHz frequencies. However, the table in the guidelines specifically recommends calculating hearing thresholds only at 0.5, 1, and 2 kHz. The authors have to clarify this.

Response: Thank you for bringing this to our attention. We have clarified that we used a modified version of the WHO guidelines, on Page 6 line 180, by stating: “All participants had clinically normal hearing, defined as an average of 25 dB HL or less in the better ear across the five frequencies tested (criteria adapted from 30,31) apart from one younger adult whose audiometric data did not save due to technical issues but this participant self-reported normal hearing.”

We decided to include 3 and 4 kHz to provide a more comprehensive representation of the speech frequency range compared to the WHO criteria (Hughes, G. W., & Halle, M. (1956). Spectral properties of fricative consonants. Journal of the Acoustical Society of America. https://doi.org/10.1121/1.1908271), which is focused more on clinical applications of this criteria. We would also like to note that if calculations were done excluding the 3 kHz frequency, all participants would still be considered to have clinically normal hearing (i.e., PTA less than or equal to 25 dB HL in the better ear).

1.4. The authors do not clearly state the calibration procedure which is an important part of this manuscript if someone wants to replicate it. Lines 284-285: What sound pressure levels were measured? To determine the sound pressure level, what type of sound was used? Was it the target sentence or the background noise? How many times did these calibrations were conducted?

Response: The sound pressure levels were measured for the CST babble and the DriverLab environmental and vehicle noises (e.g., mixture of engine noises, tires rumbling on the pavement, A/C fans). The duration of a sample was 8 minutes (the time it took for a driving scenario). These measurements were conducted on 3 samples to obtain the average sound level (dBA) reported in the manuscript. We have further clarified this procedure in the manuscript on Pages 11 and 12, lines 300-312, which now states: “The CST was presented through all DriverLab speakers. The sound levels were measured using the REED R8050 SLM and were, on average, consistent across drives. Specifically, background noise, including the CST babble and DriverLab environmental and vehicle noises (e.g., engine noises, tires rumbling on the pavement, A/C fans) were measured three times, with an average of 64 dB A across all conditions. Listening difficulty was manipulated by adjusting the target female talker relative to the background noise to obtain two signal-to-noise ratios (SNR): an SNR of +4 dB (intended to be easier listening) and an SNR of 0 dB (intended to be harder listening).”

Information about the calibration procedure of the SLM was included in the previous round of revisions and can be found in the Apparatus section of the Methods (Page 11, lines 284-286). To clarify, the SLM was calibrated before each use.

1.5. I think it is appropriate to report SNR in dB throughout the manuscript, consistently.

Response: We have added dB to all mentions of SNR throughout the manuscript.

1.6. In the previous revision, I asked about the piloting process. How was it conducted, and how many participants were involved? Additionally, how many participants participated in both the pilot and the main study? If some participants were involved in both, they would have been familiar with the study. Do the authors believe that this familiarity with the study and the tasks could have influenced the results?

Response: We now recognize that the wording describing the piloting procedure may have been misleading. The pilot participants were not a subset of the main study sample, but rather entirely different participants who never participated in the main study. Before launching the main study, piloting of the procedure and stimuli was conducted on 6 younger adults and 4 older adults. Thus, there were no familiarity or carry over effects between piloting and the main study. We have further clarified this in the manuscript, Page 12 lines 314-318, which now states: “Therefore, piloting of the experimental task, using the protocol described above with a small sample of younger (N=6) and older (N=4) adults, was conducted to ensure that the 0 dB SNR yielded poorer accuracy scores on the CST than the +4 dB SNR in both age groups, without reaching floor or ceiling effects. Participants involved in the piloting phase did not participate in the main study.”

1.7. The authors have responded to my question, stating that "it is not necessary to apply a p-value correction for multiple comparisons when the data under evaluation are not random numbers but actual observations of nature" (Rothman K. J., 1990, Epidemiology). However, I find their response regarding p-value adjustments for multiple comparisons unconvincing. If the authors do not support adjusting the p-value, they should refrain from using it altogether in this manuscript. The authors have applied a Bonferroni correction in the manuscript (lines 378-379), but this is not the only available method. The False Discovery Rate (FDR) is another appropriate approach the authors could consider.

Response: Thank you for bringing this to our attention and for offering an alternative approach. We agree that it is often beneficial to apply consistent parameters and statistical criteria throughout the results section. We also recognize that supporting the use of multiple comparison corrections for some analyses while not implementing them for others may appear inconsistent. However, we would like to clarify that different analyses in this manuscript serve distinct purposes and, therefore, require different strategies. Specifically, the ANOVAs conducted on listening and driving performance are the primary outcome measures, driven by our hypotheses and the main focus of the paper. As such, we opted for a more conservative approach by applying the Bonferroni correction for multiple comparisons, which helps reduce the likelihood of Type I errors (i.e., detecting an effect that does not actually exist). On the other hand, the correlations between listening and driving performance and baseline measures of sensory and cognitive abilities are exploratory in nature and secondary to the primary goals of the study. Since these correlations were not hypothesis-driven, and there were many comparisons, we chose to transparently implement a more liberal approach by not applying a correction for multiple comparisons. This strategy reduces the likelihood of Type II errors (i.e., failing to detect a true effect). If we had applied the Bonferroni correction to these exploratory correlations, the required p-value to detect an effect would have been extremely small (approximately 0.00083) due to the large number of comparisons being conducted.

While the False Discovery Rate (FDR) approach allows for more discoveries than stricter corrections like Bonferroni, it still involves a threshold that could prevent the detection of significant effects, especially when those effects are subtle. We hope that these exploratory analyses will help inform future research that can apply more stringent criteria to directly test these effects. Moreover, since we have a wealth of baseline data that can be analyzed in the context of the dual-task paradigms used in the current study, these exploratory analyses were intended to maximize the use of this rich dataset to gain deeper insights into the relationship between sensory and cognitive measures and our primary outcome measures. Additionally, in the discussion section, we reference Table 1, which uses t-tests to examine group differences in baseline measures (e.g., Independent Samples t-tests which found significant differences between younger and older adults for PTA and CDTT SRT). This helps provide additional evidence and context to support the exploratory correlation results we report using a more conservative approach.

We also explicitly explain to readers that these analyses are exploratory, and no corrections for multiple comparisons were applied with full transparency, indicating that these results should be interpreted with caution. We have added statements in the Discussion on Page 28, lines 894-896, and Page 31, lines 959-961, clarifying this point: “It is important to acknowledge that these associations are exploratory in nature, and no corrections for multiple comparisons were applied. Consequently, these findings should be interpreted with caution.”

1.8. The results section remains difficult to follow. The authors should focus on presenting the key findings that they want the readers to take away, ensuring clarity and emphasis on the most important messages.

Response: We acknowledge that the results section was previously lengthy and potentially challenging to navigate, as much of the findings presented in the raw values were consistent with those reported in the proportional dual-task costs. To improve the overall flow and clarity of the manuscript, we have moved the detailed analyses and figures for proportional dual-task costs to the Supplementary Materials. As a result, the results section now includes five figures instead of seven, making it more concise and easier for readers to focus on the key findings.

We still defend including the proportional dual-task costs in the Supplemental Materials rather than removing them all together given that this is a very common approach typically expected for dual-task studies and most importantly because they do offer additional insights. Specifically, proportional dual-task costs provide unique information regarding whether the differences observed between single- and dual-task conditions are significantly greater for one group compared to the other. While the raw values offer valuable information, they do not directly address the magnitude of these differences between groups—an essential finding we aim to highlight. To ensure these unique findings are accessible to readers, we have included a brief paragraph at the end of the Listening Performance – Accuracy and Driving Performance – Standard deviation of lane position (SDLP) sections referencing the Supplementary Materials, summarizing the key unique insights from the proportional dual-task costs without reporting the full analyses. Lastly, determining task prioritization/relative task costs to listening and driving performance is only possible by referring to the proportional dual-task costs.

The Results Section is now presented as:

Listening Performance – Accuracy

• Correlations (listening accuracy)

Driving Performance – Standard deviation of lane position (SDLP)

• Correlations (standard deviation of lane position)

Relative Costs to Listening and Driving Performance

• Comparison between listening and driving performance proportional dual-task costs.

2. Responses to Reviewer #1 Comments

2.1. I still find this manuscript lengthy and challenging to follow. The extensive analyses, lengthy results section, and inclusion of 11 figures make it a difficu

---

## [Decision Letter · Decision Letter 2]

13 Apr 2025

PONE-D-24-41749R2Dual-task costs of listening while driving in older and younger adultsPLOS ONE

Dear Dr. Bak,

Thank you for submitting your manuscript to PLOS ONE. After careful consideration, we feel that it has merit but does not fully meet PLOS ONE’s publication criteria as it currently stands. Therefore, we invite you to submit a revised version of the manuscript that addresses the points raised during the review process.

We look forward to receiving your revised manuscript.

Kind regards,

Sangamanatha Ankmnal Veeranna, Ph.D.

Academic Editor

PLOS ONE

**Journal Requirements:**

Reviewers' comments:

Reviewer's Responses to Questions

**Comments to the Author**

1. If the authors have adequately addressed your comments raised in a previous round of review and you feel that this manuscript is now acceptable for publication, you may indicate that here to bypass the “Comments to the Author” section, enter your conflict of interest statement in the “Confidential to Editor” section, and submit your "Accept" recommendation.

Reviewer #1: (No Response)

2. Is the manuscript technically sound, and do the data support the conclusions?

Reviewer #1: Yes

3. Has the statistical analysis been performed appropriately and rigorously? 

Reviewer #1: Yes

4. Have the authors made all data underlying the findings in their manuscript fully available?

Reviewer #1: Yes

5. Is the manuscript presented in an intelligible fashion and written in standard English?

Reviewer #1: Yes

6. Review Comments to the Author

**Reviewer #1: ** Thank you to the authors for addressing the main suggestions. The manuscript now reads more clearly and is easier to follow. However, I have a few minor suggestions—particularly concerning the discussion section—that I recommend the authors consider before the manuscript is ready for publication:

Line 675: There is a typographical error in the word "adults."

Lines 676–679: Please provide a clearer and more valid explanation for this observation.

Lines 682–683: The sentence "There were also some (albeit inconsistent) indicators of associations between word recognition accuracy and poorer performance" is unclear. It is not evident whether an association was found. Please specify the exact findings more clearly.

Lines 730–733: I recommend the authors elaborate further on this point. In the listening effort literature, pure-tone average (PTA) is often found to be a poor indicator of listening effort and disability when compared to measures such as speech-in-noise performance and self-report tools (see Alhanbali et al., 2018).

Lines 767–789: I suggest either removing this section or substantially shortening it. It is currently difficult to follow, and its relevance to the rest of the discussion is unclear.

Lines 832–838: The argument presented here is not entirely convincing. Although older adults may have more driving experience, age-related declines in cognitive, motor, and other functions are likely to offset the benefits of experience.

Conclusion section: The recommendation for hearing aid use does not seem appropriate in the context of this study, as the participants all had normal hearing.

7. PLOS authors have the option to publish the peer review history of their article (what does this mean? ). If published, this will include your full peer review and any attached files.

**Do you want your identity to be public for this peer review?** For information about this choice, including consent withdrawal, please see our Privacy Policy .

Reviewer #1: **Yes: ** Sara Alhanbali

---

## [Author Response · Author response to Decision Letter 3]

21 Apr 2025

1. Responses to Reviewer #1 Comments

1.1. Line 675: There is a typographical error in the word "adults."

Response: Thank you - we have corrected the error in the revised manuscript.

1.2. Lines 676–679: Please provide a clearer and more valid explanation for this observation.

Response: Thank you for pointing out the ambiguity in our description of the study findings. We have revised the manuscript to present the word recognition accuracy results more clearly. Page 28, lines 695-700, now states: “The current study found that younger adults’ word recognition accuracy was similar across conditions and was not affected by driving difficulty or dual-tasking. However, for older adults, word recognition accuracy was poorer in the more difficult driving condition (i.e., City) compared to the listening only condition, regardless of listening difficulty (i.e., +4 dB SNR and 0 dB SNR). Further, older adults’ word recognition accuracy was also poorer during the easier driving condition (i.e., Rural) compared to the listening only condition when listening was less difficult.”

1.3. Lines 682–683: The sentence "There were also some (albeit inconsistent) indicators of associations between word recognition accuracy and poorer performance" is unclear. It is not evident whether an association was found. Please specify the exact findings more clearly.

Response: This sentence was meant to convey that associations among word recognition accuracy and other sensory and cognitive measures were found only for some dual-task conditions, but importantly, not in any single-task conditions. We had added “albeit inconsistent” because these associations were not found for every dual-task condition, but we understand that this may have caused confusion. To convey this more clearly, we have revised the sentence on Page 28, lines 702–706, which now states: “There were also some associations between word recognition accuracy and poorer performance on the other sensory and cognitive measures (e.g., PTA, MoCA, Trail Making, Digit Span, UFOV) under some dual-task conditions, but no associations were observed for any single-task conditions (see Figure 6).”

1.4. Lines 730–733: I recommend the authors elaborate further on this point. In the listening effort literature, pure-tone average (PTA) is often found to be a poor indicator of listening effort and disability when compared to measures such as speech-in-noise performance and self-report tools (see Alhanbali et al., 2018).

Response: Thank you for this comment. We agree that pure-tone average (PTA) may not be a reliable indicator of listening difficulties experienced by older adults under noisy listening conditions. This is why we differentiated the associations of CDTT SRT with driving performance from the associations between PTA and driving performance. In response, we have expanded this discussion in the manuscript by more explicitly linking previous literature to the present findings. Page 30, lines 762-771, now states: “For example, better speech-in-noise thresholds (i.e., CDTT SRT) were associated with better driving performance across almost all driving conditions. Notably, audiometric thresholds (i.e., PTAs) were not associated with driving performance under any condition. PTAs may not be a reliable indicator of speech understanding abilities in noisy environments, as older adults with clinically normal audiometric pure-tone thresholds often still perform more poorly on speech-in-noise tests than younger adults with normal hearing (Dubno 1984; Fullgrabe 2015). The current findings appear to extend prior findings to more complex driving tasks, whereby better speech-in-noise thresholds rather than audibility (lover PTA) are associated with better management of the relative demands of performing a speech-in-noise word recognition task while driving.”

1.5. Lines 767–789: I suggest either removing this section or substantially shortening it. It is currently difficult to follow, and its relevance to the rest of the discussion is unclear.

Response: Thank you for this suggestion. The final paragraph originally addressed how discrepancies in previous literature may have potentially resulted from differences in driving task complexity. The preceding paragraph focused on similar discrepancies, but in relation to listening task differences. It also used current findings on associations between sensory and cognitive measures and driving performance to help interpret prior research. Upon further reflection, we have combined these two paragraphs into a single, more concise discussion that integrates both points and streamlines the content. We hope these changes make this section easier to follow, and that the relevance to the rest of the discussion is clearer. Please see Pages 31-32, lines 790-825.

1.6. Lines 832–838: The argument presented here is not entirely convincing. Although older adults may have more driving experience, age-related declines in cognitive, motor, and other functions are likely to offset the benefits of experience.

Response: While the main objective of the study was to evaluate age-related differences in dual-task driving performance, driving history and experience is essentially confounded with age and age-related sensory/motor/cognitive changes. Specifically, older adults in our study had, on average, over 49 years of driving experience, compared to only 9 years of driving experience in younger adults. It is very likely that both age-related declines in sensory/cognitive/motor functioning and experience both contributed to the driving outcomes observed in this study, with the relative contributions of each remaining unknown, but acting in opposite directions. Also notable is that older adults in our study were also very active, healthy, participants who were screened for any significant sensory or cognitive declines or impairments. As such, the negative effects of any milder age-related declines may have been less impactful than what might be expected from the more general older adult population who have various impairments and comorbidities (as we already address as a limitation). As such, it might be expected that such an extensive history of driving experience combined with generally mild declines, may have allowed experience to play a compensatory role, particularly in the easier driving conditions (i.e. Rural), for which we indeed saw no age-related differences in the results. We have thus adjusted this section to address these considerations more directly, which now reads on Page 34, lines 906-911: “The older adults in this study had significantly more years of driving experience (M= 49.05 years) than younger adults (M=9.88 years). Age-related differences in driving experience between age groups in the current study may have reduced the age-related effects on some of our outcome measures, particularly under the easier driving conditions (Rural) because older adults could compensate for potential age-related declines in sensory/cognitive/motor abilities by relying on well-practiced driving skills and expertise.”

1.7. Conclusion section: The recommendation for hearing aid use does not seem appropriate in the context of this study, as the participants all had normal hearing.

Response: Thank you for this note. We have removed the mention of hearing aids from the sentence. Pages 35-36, lines 946-951, now states: “Future research could examine ways to potentially mitigate these dual-task effects by, for example, working memory training to help reduce the cognitive load associated with speech understanding during driving, optimization of the vehicle acoustics or use of technology to improve the SNR, or minimization of auditory distractions, especially when driving situations become more complex (e.g., in the City).”

---

## [Editor Report · Decision Letter 3]

29 Apr 2025

Dual-task costs of listening while driving in older and younger adults

PONE-D-24-41749R3

Dear Dr. Bak,

We’re pleased to inform you that your manuscript has been judged scientifically suitable for publication and will be formally accepted for publication once it meets all outstanding technical requirements.

Kind regards,

Sangamanatha Ankmnal Veeranna, Ph.D.

Academic Editor

PLOS ONE

---

## [Editor Report · Acceptance letter]

PONE-D-24-41749R3

PLOS ONE

Dear Dr. Bak,

I'm pleased to inform you that your manuscript has been deemed suitable for publication in PLOS ONE. Congratulations! Your manuscript is now being handed over to our production team.

Kind regards,

on behalf of

Dr. Sangamanatha Ankmnal Veeranna

Academic Editor

PLOS ONE